


# Turbulence-permitting air pollution simulation for the Stuttgart metropolitan area

Thomas Schwitalla*[1], Hans-Stefan Bauer[1], Kirsten Warrach-Sagi[1], Thomas Bönisch[2], Volker Wulfmeyer[1]

[1] Institute of Physics and Meteorology, University of Hohenheim, Garbenstrasse 30, 70599 Stuttgart, Germany.

[2] High-Performance Computing Center Stuttgart (HLRS), Nobelstrasse 19, 70569 Stuttgart, Germany

*Correspondence to*: Thomas Schwitalla (thomas.schwitalla@uni-hohenheim.de)

**Abstract.** Air pollution is one of the major challenges in urban areas. It can have a major impact on human health and society and is currently a subject of several litigations at European courts. Information on the level of air pollution is based on near surface measurements, which are often irregularly distributed along the main traffic roads and provide almost no information about the residential areas and office districts in the cities. To further enhance the process understanding and give scientific support to decision makers, we developed a prototype for an air quality forecasting system (AQFS) within the EU demonstration project "Open Forecast".

For AQFS, the Weather Research and Forecasting model together with its coupled chemistry component (WRF-Chem) is applied for the Stuttgart metropolitan area in Germany. Three model domains from 1.25 km down to a turbulence permitting resolution of 50 m were used and a single layer urban canopy model was active in all domains. As demonstration case study the 21 January 2019 was selected which was a heavy polluted day with observed $PM_{10}$ concentrations exceeding 50 µg m$^{-3}$.

Our results show that the model is capable to reasonably simulate the diurnal cycle of surface fluxes and 2-m temperatures as well as evolution of the stable and shallow boundary layer typically occurring in wintertime in Stuttgart. The simulated fields of particulates with a diameter of less than 10 µm ($PM_{10}$) and Nitrogen dioxide ($NO_2$) allow a clear statement about the most heavily polluted areas apart from the irregularly distributed measurement sites. Together with information about the vertical distribution of $PM_{10}$ and $NO_2$ from the model, AQFS will serve as a valuable tool for air quality forecast and has the potential of being applied to other cities around the world.

## 1. Introduction

Currently more than 50 % of the global population live in cities whereas the United Nations (UN) expect a further increase by about 10 % in 2030 (UN, 2018). The UN also expect that in 2030 34% of the world population will reside in cities with more than 500 000 inhabitants.

To protect human life, the World Health Organization (WHO) proposed maximum permittable pollution levels (Maynard et al., 2017 and references therein). E.g. for particulate matter with particle diameters less than 10 µm ($PM_{10}$), the critical value is an annual mean concentration of 20 µg m$^{-3}$ or a daily mean value of 50 µg m$^{-3}$ (WHO, 2005). For Nitrogen dioxide ($NO_2$) the critical values are 200 µg m$^{-3}$ and 40 µg m$^{-3}$ as daily and annual mean values, respectively.



Due to a strong increase of road traffic in major European cities (Thunis et al., 2017), these pollution levels are
often violated in larger cities. This can lead to health and environmental problems and is currently part of several
litigations e.g. at the German Federal Administrative Court dealing with possible driving bans for non low-
emission vehicles. The basis for these litigations are mostly few local observations which are unevenly distributed.
In combination with special meteorological conditions like winter time thermal inversion layers it can be
misleading to conclude about the overall air quality in the city. According to e.g. the German Federal Immission
Control Ordinance[1] it is sufficient that traffic related measurements are representative for a section of 100 m, but
this is not representative for the commercial and office districts in the cities that are suffering from traffic control
in case of fine dust alerts and residential areas. Namely in residential areas health protection action plans require
representative air quality measures.
Therefore it becomes important to apply a more scientifically valid approach by applying coupled atmospheric
and chemistry models to predict air quality. Regional and global atmospheric models like the Weather Research
and Forecasting (WRF) model  (Skamarock et al., 2019), the Consortium for Small Scale modeling (COSMO;
Baldauf et al., 2011),  the Icosahedric Nonhydrostatic model (ICON; Zängl et al., 2015), or the Regional Climate
Model system (RegCM4; Giorgi et al., 2012) are often used to force offline chemistry transport models like
CHIMERE (Mailler et al., 2017), LOTOS-EUROS (Manders et al., 2017), EURopean Air Pollution Dispersion
(EURAD; Memmesheimer et al., 2004), and Model for OZone And Related chemical Tracers (MOZART)
(Brasseur et al., 1998; Horowitz et al., 2003).
Several studies showed that combining an atmospheric model with an online coupled chemistry component is a
suitable tool for air quality and pollution modeling in urban areas at the convection permitting  (CP) resolution
(Fallmann et al., 2014; Kuik et al., 2016; Zhong et al., 2016; Kuik et al., 2018; Huszar et al., 2020) .
Compared to chemical transport models,  coupled models like WRF-Chem (Grell et al., 2005), COSMO-ART
(Vogel et al., 2009),  ICON-ART (Rieger et al., 2015), and the Integrated Forecasting System (IFS) MOZART
(Flemming et al., 2015) allow for a direct interaction of aerosols with radiation leading to a better representation
of the energy balance closure at the surface as it would be the case when applying an offline chemistry model.
As usually the terrain and land cover over urban areas show fine scale structures which are not resolved even by a
CP resolution, there is a need for turbulence permitting (TP) simulations with horizontal grid increments of a few
hundred meters or even less. Important features are, e.g., urban heat island effects (Fallmann et al., 2014; Fallmann
et al., 2016; García-Díez et al., 2016; Li et al., 2019) and local wind systems like mountain and valley winds due
to differential heating (Corsmeier et al., 2011; e.g. Jin et al., 2016). Also, micro- and mesoscale wind systems can
develop due to urban structures and the heterogeneity of the land surface. It is well known that TP simulations are
a promising tool to further enhance the understanding of processes in the atmospheric boundary layer (Heinze et
al., 2017b; Panosetti et al., 2016; Heinze et al., 2017a; Bauer et al., 2020) in urban areas (Nakayama et al., 2012;
Maronga et al., 2019; Maronga et al., 2020).
In order to further enhance the quality of  the simulations, building and urban canopy models (UCM) are developed
(Martilli et al., 2002; Kusaka and Kimura, 2004; Salamanca and Martilli, 2010; Maronga et al., 2019; Scherer et
al., 2019; Teixeira et al., 2019).  The main purpose of UCMs is to provide a better description of the lower

---

[1] https://www.gesetze-im-internet.de/bimschv_39/anlage_3.html





boundaries over urban areas such as building, roof and road geometries and their interactions with atmospheric
water vapor, wind, and radiation.
With the EU-funded project Open Forecast (https://open-forecast.eu/en/) it was intended to develop a prototype
for an air quality forecasting system (AQFS) for the Stuttgart metropolitan area in southwest Germany. Open
Forecast is a demonstration project to show the potential of open data combined with supercomputer resources to
create new data products for European citizens and public authorities. The long-term goal is to provide end users
and political decision-makers a useful tool, particularly considering further urbanization, heat island effects as well
as potential driving restrictions due to recent EU decisions on emission limits.
For our AQFS we use the WRF-Chem NWP model (Grell et al., 2005; Skamarock et al., 2019) as the WRF model
is extensively evaluated over Europe at different time scales and horizontal resolutions (San José et al., 2013;
Warrach-Sagi et al., 2013; Milovac et al., 2016; Lian et al., 2018; Molnár et al., 2019; Bauer et al., 2020; Coppola
et al., 2020; Schwitalla et al., 2020). It can easily be set up in a nested configuration over all regions of the Earth.
Compared to PALM-4U model, the nested model domains are driven by the full atmospheric and chemical
information from the parent domain along its lateral boundaries. Also, it contains well-characterized combinations
of parameterizations of turbulence and cloud microphysics in the outer domain that are consistent with the inner
TP domains where the high-quality cloud parameterization remains. No switch between different model systems
is required, which is expected to provide a great advantage with respect to the skill of air pollution and
meteorological forecasts.
To enhance the forecast skill, suitable variational and ensemble-based data assimilation systems are already in
place to further improve the meteorological initial conditions (Barker et al., 2012; Zhang et al., 2014; Kawabata et
al., 2018; Thundathil et al., 2020) and the chemical initial conditions (Chen et al., 2019; Sun et al., 2020) but this
is beyond the scope of our study.
The Parallelized Large-Eddy Simulation Model (PALM) model (Maronga et al., 2015) is another widely used TP
simulation model over Europe. PALM did not include the full interaction between land-surface, radiation, cloud
microphysics and chemistry during the performance of our study. The very recent version 6.0 of PALM-4U
(PALM for urban applications) (Maronga et al., 2020) is expected to contain a fully coupled chemistry module
(Khan et al., 2020, under review).
Fallmann et al. (2016) and Kuik et al. (2016) performed air quality simulations with WRF-Chem over the cities of
Berlin and Stuttgart on a CP resolution down to 1km and less than 40 model levels. They used the TNO-MACC
emission inventory (Kuenen et al., 2014) which is available as an annual totals on a 7 km x 7 km resolution. As
the topography of Stuttgart is very complex, the AQFS applies the WRF-Chem model on a turbulence permitting
horizontal resolution using 100 model levels to account for the shallow boundary layer occurring during
wintertime. In addition, we applied a local emission data set from the Baden-Württemberg State Institute for the
Environment, Survey and Nature Conservation available as annual mean on a horizontal resolution of 500 m x 500
m to resolve fine-scale emission structures.
Our study focuses on the methodology how to set up a AQFS prototype by using WRF-Chem and its application
to a typical wintertime situation in the Stuttgart metropolitan area. The manuscript is set up as follows: section 2
describes the design of our AQFS model system on the turbulence permitting resolution of 50 m followed by a



description of the selected case study. Section 4 shows the results including a discussion, sect. 5 summarizes our
work and gives an outlook on potential future enhancements of the AQFS prototype.
**2.  AQFS design**
**2.1.  WRF model set-up**
For our AQFS, we selected the Advanced Research WRF-Chem model in version 4.0.3 (Grell et al., 2005;
Skamarock et al., 2019). To reach the targeted resolution of 50 m, three model domains have been applied with
horizontal resolutions of 1250 m, 250 m, and 50 m and encompasses 800*800 grid cells in the outer domain and
601*601  grid cells in the two inner TP domains. The reason to start with a resolution of 1250 m in the outermost
domain is to avoid the application of a convection parametrization which can deteriorate the model results (Prein
et al., 2015; Coppola et al., 2020). The areas of model domain 1 and 3 are shown in Fig. 1.
As seen from Fig. 1b, the Stuttgart metropolitan area is characterized by an elevation variation of more than 300
m. The lowest elevation is approx. 220 m in the basin while the highest elevation reaches up to 570 m. As the main
traffic roads are in the basin, especially during wintertime this often leads to a worsening of the air quality as the
surrounding prevents an air mass exchange due to the stationary temperature inversion.
For the WRF model system land cover and soil texture fields are not available at resolutions higher than 500m.
Therefore we reclassified land cover data from the Copernicus CLC 2012 data set (European Union, 2012),
available on a resolution of 100 m,  from the original 44 categories to the categories applied in the WRF model for
the simulations of the outer 2 domains. For the innermost model domain, we incorporated the most recent high-
resolution land-cover data set from the Baden-Württemberg State Institute for the Environment (LUBW), which
is derived from Landsat (Butcher et al., 2019) in 2010 and is available at 30 m resolution (https://udo.lubw.baden-
wuerttemberg.de/public/) This data set was also reclassified to the corresponding land cover categories used in
WRF and is shown in Fig. 2.
The resolution of the provided default Food and Agriculture Organization of the United Nations (FAO) soil texture
data is only 10 km, therefore we used soil texture data from the International Soil Reference and Information
Centre (ISRIC) SoilGrids project (Hengl et al., 2014; Hengl et al., 2015). These data are available on a resolution
of 250 m. Terrain information was provided by the National Center for Atmospheric Research (NCAR) derived
from the Global multi-resolution terrain elevation data 2010 (GMTED2010) data set (Danielson and Gesch, 2011)
for domain 1.  As the horizontal resolution of the GMTED2010 data set is 1 km, the 3" gap-filled Shuttle Radar
Topography Mission (SRTM) data set (Farr et al., 2007) is used for domain 2. As this resolution is still too coarse
for our targeted resolution of 50 m, the Digital Elevation model Europe (EU-DEM; European Union, 2017),
available at a  resolution of 25 m, is used for the innermost domain.
In our set-up, we use 100 vertical levels for all domains using the traditional terrain following coordinate system
in WRF; 20 of the levels are distributed in the lowest 1100 m above ground level (AGL). All domains apply the
Noah-MP land surface model (Niu et al., 2011; Yang et al., 2011), the revised MM5 surface layer scheme based
on Monin-Obukhov similarity theory (Jiménez et al., 2012), the Thompson 2-moment cloud microphysics scheme
(Thompson et al., 2008) and the Rapid Radiative Transfer Model for GCMs (RRTMG; Iacono et al., 2008) for
parametrizing longwave and shortwave radiation. Due to the coarser resolution of the outermost domain, we
applied the Yonsei University (YSU; Hong et al., 2006) planetary boundary layer (PBL) parametrization in D01



only. As suggested by the WRF user guide, we applied the sub-grid turbulent stress option for momentum
(Kosovic, 1997) in domains two and three. The complete namelist settings are provided in the supplement.
As the finest resolution applied for the AQFS is 50 m, the more sophisticated Building Effect Parameterization
(BEP; Martilli et al., 2002) is not applied. as this scheme does not work with our selection of parametrizations.
Instead, the single layer urban canopy model (UCM) (Kusaka and Kimura, 2004) is selected to improve the
representation of the urban canopy layer and the surface fluxes. The parameters needed by the UCM are read in
from the lookup table URBPARAM.TBL which was adjusted for the Stuttgart area following Fallmann (2014).
Atmospheric chemistry is parametrized  by the Regional Acid Deposition Model 2nd generation (RADM2) model
(Stockwell et al., 1990). RADM2 features 63 chemical species including photolysis and more than reactions.
Aerosols are represented by the Modal Aerosol Dynamics Model for Europe (MADE) and Secondary Organic
Aerosol Model (SORGAM) scheme (Ackermann et al., 1998; Schell et al., 2001) considering size distributions,
nucleation, coagulation, and condensational growth. The combination of RADM2_MADE-SORGAM is a
computationally efficient approach and is widely used for simulations over Europe (Forkel et al., 2015; Mar et al.,
2016). To further enhance vertical mixing of CO to higher altitudes during nighttime over urban grid cells, the if-
statements in the  dry deposition driver of WRF-Chem  at lines 690 and 707 have been deleted according as shown
in the supplement of Kuik et al. (2018).
Due to the complexity of the chemistry model in combination with the very high horizontal resolution and the
calm meteorological conditions, the adaptive model time step option was chosen instead of a fixed time step.
Model output is available in 5 min intervals for the innermost model domain.
Our single day case study on the turbulence permitting (TP) scale is designed to serve as a test bed to set up an air
quality forecasting system prototype for the Stuttgart metropolitan area. For process studies, the model chain itself
can be applied to other areas over the globe as long as 1)  detailed land cover and soil texture data are available,
2) high-resolution emission data not only from traffic are available. The new model system can be even applied in
a forecast and warning mode, if near real time emission data exist. As the computational demands of applying
WRF-Chem on the TP scale are very high, access to an HPC system is a prerequisite.
**2.2. Model initialization**
The meteorological initial and boundary conditions were provided by the operational ECMWF integrated
forecasting system (IFS) analysis on model levels. The IFS is a global model with 9 km horizontal resolution and
applies a sophisticated four-dimensional variational (4DVAR) data assimilation system (Bonavita et al., 2016).
The data have been retrieved from the ECMWF Meteorological Archival and Retrieval System (MARS) and were
interpolated to a resolution of 0.05°.
The initialization and provision of the boundary conditions of the chemistry of the model is done with data from
the Whole Atmosphere Community Climate Model (WACCM; Marsh et al., 2013) using the Model for Ozone and
Related Chemical Tracers (MOZART) conversion tool MOZBC (Pfister et al., 2011). As the resolution of
WACCM is very coarse, the input data was  extended by the ECMWF Copernicus Atmosphere Monitoring Service
(CAMS) reanalysis data set on 60 model levels and 40 km horizontal resolution (Inness et al., 2019).
**2.3. Emission data**





The emission data set used in this study is a combination of three products.  Global input data sets containing
coarse resolution emissions from different sources are obtained from the BRAMS numerical modeling system
(Freitas et al., 2017).The PREP-CHEM-SRC tool (Freitas et al., 2011) is then applied as pre-processor to convert
these emissions to the appropriate  WRF units and interpolate the data onto the  WRF model grid.
As global emission data sets have a very coarse resolution in space and time, higher resolution emission data for
Europe from the Copernicus Atmosphere Monitoring Service (CAMS; Copernicus) CAMS-REG-AP product
became available (Granier et al., 2019). Its resolution is approx. 7x7 km and it is based on total annual emissions
from 2016. This product provides emissions of $PM_{10}$, $PM_{2.5}$, $SO_2$, CO, $NO_x$, and $CH_4$ and contains sources from
different sectors, separated into ten different categories following the Gridded Nomenclature For Reporting
(GNFR; Granier et al., 2019).
The third emission data set (BW-EMISS) deployed in our study was obtained from the Baden-Württemberg State
Institute for the Environment (LUBW). This data set contains annual mean emissions from different sectors
following the GNFR classification and is currently available only until 2014 and has a horizontal resolution of 500
m. Unfortunately, more recent quality-controlled data sets were not available when our study was performed. It is
expected that annual emissions for 2018 will become available by mid of 2021.
As CAMS-REG-AP and BW-EMISS only contain annual sums or annual mean values, a temporal decomposition
was applied for both data sets following Denier van der Gon et al. (2011). Depending on the GNFR code, the data
are first projected onto the corresponding month, followed by the corresponding day of the week and the hour of
the day. A similar approach was performed e.g. in Resler et al. (2020, under review) for the city of Prague.  After
finishing the decomposition, the data are converted to the corresponding units and interpolated onto the WRF
model grid using the Earth System Model Framework (ESMF; Valcke et al., 2012) interpolation utilities.
Figure 3 shows an example of the $NO_2$ emissions derived from the CAMS-REG-AP product (left) and the emission
data derived from the LUBW data set (right) on January 21, 2019 at 07 UTC.
Due to its much higher horizontal resolution, the BW-EMISS data set (Fig. 3b) shows much more detailed
structures for the $NO_2$ emissions which are mainly caused by road traffic.
In addition, the following adjustments have been performed: 1) $NO_x$ emissions from forest grid cells have been
reduced by 90 %, 2) Road traffic $NO_x$ emissions were transformed into 90 % NO and 10 % $NO_2$ emissions
following Kuik et al. (2018) 3) All emissions from Stuttgart airport were reduced by 90 % during the nighttime
flight ban between 00 UTC and 04 UTC as well as after 21 UTC.
The WRF-Chem model only ingests one emission data set per species, hence emissions from the different GNFR
categories have been accumulated to a single emission data set before performing the simulation. Figure 4
summarizes all necessary steps and the complete data and workflow of the AQFS prototype.
**2.4. Observations**
We used data from three meteorological stations (Stuttgart-Schnarrenberg ( 48.8281°N 9.2°E, elevation 314 m),
Stuttgart Airport (48.6883°N 9.2235°E, elevation 375 m),  and Institute of Physics and Meteorology (IPM) at the
University of Hohenheim (48.716°N 9.213°E, elevation 407 m) to validate the simulated 2-m temperatures; data





are available every 10 minutes. The locations are indicated by the black dots Fig. 1b. In addition, the radiosonde
data from Stuttgart-Schnarrenberg were used.
As the incorporated emissions are from 2014 and are based on annual values, it would be rather misleading to
compare the observed pollutant concentrations directly with the model output. For instance, the actual traffic, the
sequence of traffic lights and traffic congestions of this particular day cannot be realistically represented. In
addition, all diagnosed or prognostic chemical quantities are only available on model levels (with the lowest model
half level being at ~15 m above ground) so that simulated concentrations need to be interpolated to the
measurement heights at 2.5—3.5 m AGL. This extrapolation may cause even more uncertainty. Therefore we
decided not to use pollution measurements for model evaluation but we studied the results based on process
understanding and plausibility arguments.
**3. Case study description**
For our study, we selected 21 January 2019. This day was characterized as "fine dust alarm" situation (Stuttgart
Municipality and German Meteorological Service (DWD), 2019) which is defined by a combination of the
following criteria:
1. Expected daily maximum $PM_{10}$ concentration at Stuttgart Neckartor (NT in Fig. 1b) is higher than 30 µg

$m^{-3}$

2. No rain on the following day
3. 10-m wind speed less than 3 m s$^{-1}$ from south to northwest directions (180-330 °)
4. Nocturnal atmospheric inversion
5. Mixing layer depth less than 500 m during the day
6. Daily average 10-m wind speed less than 3 m s$^{-1}$ from all directions
A sufficient criterion is a higher $PM_{10}$ concentration following (1). If (1) is not fulfilled, then (2) and (3) together
with either (4) and/or (5) must be fulfilled. If only (4) or (5) is fulfilled, then (6) must be considered. For our case
study, the criteria 1-5 were fulfilled.
Figure 5 shows the observed $PM_{10}$ and $NO_2$ concentrations at several stations in our model domain. From Fig. 5a
the high $NO_2$ concentrations at Neckartor and Hohenheimer Strasse occurring after sunrise can be clearly
identified. While these measurements are taken next to main roads, the other stations show considerably lower
$NO_2$ concentrations throughout the day. The $PM_{10}$ concentrations (Fig. 5b) show extremely high values at
Neckartor exceeding 100 µg m$^{-3}$ around noon time and the evening rush hour which clearly meets the main criteria
of the "fine dust alarm situation". The other stations, which are not directly taken near main roads with heavy
traffic show considerably lower $PM_{10}$ concentrations around 40 µg m$^{-3}$ .
This day was a typical winter weather situation. Central Europe was located at the east flank of a blocking high
pressure system located over the East Atlantic together with moderate to low horizontal geopotential gradients and
resulting weak winds at 500 hPa in southwestern Germany (Fig. 6a).
Near surface temperatures are below freezing level, between 1000 and 850 hPa very light easterly winds
characterize the flow, and a dry layer is present around 925 hPa (Fig. 6b). Above 850 hPa, the wind direction
rapidly changes to westerly directions, but the wind speeds remain below 5 m s$^{-1}$ (see Fig. 7a).





The inversion between the two air masses inhibits vertical mixing leading to higher concentrations of aerosols in
the lowest few hundred meters above ground (AGL) and preventing air mass exchange aloft. This inversion is
further enhanced by the special orography of Stuttgart city (see later Fig. 15).
**4. Results and Discussion**
**4.1. Meteorological quantities**
A Skew-T diagram of the observed and simulated temperature, dew point, and wind profiles allows to evaluate the
stratification conditions of the model. Figure 7a shows the vertical profile of the model initial conditions at
Stuttgart-Schnarrenberg valid at 00 UTC  21 January 2019 in comparison with the observations.
The initial conditions agree well with the sounding showing a weak temperature inversion around 900 hPa with
high relative humidity values up to 650 hPa. The observed and simulated lifting condensation level is 940 hPa and
the integrated water vapor (PWAT) is 8 mm. Wind speed and direction agree with the observations showing a
wind shear above 850 hPa associated with low wind speeds of less than 5 m s$^{-1}$.
To further evaluate the stratification conditions during the day, Figure 7b shows the observed and simulated
temperature, dew point, and wind profiles at 11 UTC. The vertical structure of the observation and the simulation
has an almost perfect agreement. The temperature inversion layer at 910 hPa is well captured although the
simulated temperatures below the inversion are too high by about 1.5 K. The humidity profile (expressed as
dewpoint profile) is also very well captured with the largest moisture content below 870 hPa. Wind speed and
direction above 850 hPa agree well with the observation throughout the atmosphere. In regard of the vertical model
resolution, the wind situation in the lowest 1000 m AGL is also reasonably represented.
Figure 8 exemplarily shows the simulated 2-m temperature together with 10-m wind velocities at 12 UTC (noon
time) to display the complexity of the Stuttgart metropolitan area.
The 2-m temperatures show a daytime warming of downtown Stuttgart and the Neckar Valley while still
temperature slightly below 0°C are present at higher elevations (blue colors in Fig. 8). The wind situation is very
complex due to weak wind speeds in combination with a shallow boundary layer (see later Fig. 16) but the wind
flow along the upper Neckar river (south of 48.75°) is strongly pronounced. After sunset, wind speed starts to
increase and the channeling effect along the Neckar weakens (not shown).
Figure 9 shows an evaluation of the diurnal cycle of 2-m temperatures at the three measurement sites
Schnarrenberg, IPM and airport.  Sunrise is at 07 UTC and sunset at 16 UTC and the model data are averaged over
5 grid cells around the measurement site. The northern station Schnarrenberg shows a lower temperature
throughout the day than the other two stations, which are situated 3 km apart at a similar elevation. The temperature
is about 1 K colder during the day and 0.5 K colder during the night.
At Schnarrenberg, the observed diurnal cycle is reasonably well simulated with WRF. Between 00 and 15 UTC, a
warm temperature bias of 1 K is present in the simulation, which turns into a small negative bias after sunset. At
IPM, the simulation shows a cold bias until 04 UTC turning into a warm bias as the strong temperature drop is not
simulated until 06:30 UTC. After 09 UTC until sunset the simulated temperature agrees well with the observations
while later a cold bias of around 1 K is present.





For the airport station, the model stays too warm with a positive bias of almost 2 K between 05 and 09 UTC.
During the further course of the day, the bias reduces to 1 K at noon while after sunset it turns into a negative bias
of 1 K.
A possible reason for the larger differences at the airport and IPM before (after) sun rise (sun set) is the occurrence
of low stratus or fog. At the beginning of the simulation, cloud coverage were reported by 5—7 octas (broken
clouds) over Schnarrenberg and the airport at approx. 500 m AGL (not shown) while after 04 UTC the low level
clouds started to diminish at Schnarrenberg first leading to a strong cooling until the early morning which is seen
as a temperature drop in the observations shown in Fig. 9. This temperature drop at Schnarrenberg and IPM is also
simulated but with a delay of approx. 2 h. During the evening transition and the following night, the low stratus
is developing again at the measurement sites with a ceiling of 500 m AGL but is not simulated and thus contributes
to a stronger cooling in the model. Another contributing factor to the delayed cloud dissipation could be the
turbulence spin-up time (Kealy et al., 2019), but this is beyond the scope of this study.
Although no measurements of sensible heat and ground heat fluxes are available, diurnal cycles of the fluxes at
the three locations IPM, Schnarrenberg, and airport were investigated. Figure 10 shows the simulated surface
sensible heat and ground heat flux at the three different meteorological measurement sites.
The sensible heat flux (Fig. 10a) shows a typical diurnal cycle with fluxes around zero before (after) sunrise
(sunset). During the day, the model simulates typical wintertime sensible heat fluxes between 40 and 100 W/m²
(e.g. Zieliński et al., 2018), which nicely shows a dependency on the different underlying land cover types. Lower
sensible heat fluxes occur over the sparsely vegetated surface at the airport as compared to the cropland station
IPM. As the algorithm to diagnose the 2-m temperature in NOAHMP is rather complex, no clear correlation
between SH and the 2-m temperature shown in Fig. 9 can be made. The latent heat fluxes (not shown) are almost
zero at Schnarrenberg and less than 10 W m$^{-2}$ at the other two locations due to cold and dry winter conditions
The simulated ground heat flux (Fig. 10b) shows an interesting behavior. Until sunrise, the simulated GRDFLX at
the airport and IPM shows fluctuations around -50 W m$^{-2}$ indicating some low levels clouds in accordance with
the too high simulated 2-m temperatures shown in Fig. 9. During the further course of the day, IPM and airport
show a clear diurnal cycle with maximum values between 100 and 170 W m$^{-2}$ reflected in the highest surface
temperatures during the day (not shown).
At Schnarrenberg, most of the time the ground heat flux is less than zero indicating a cooling of the soil, while
between 12 UTC and 16 UTC small positive values are simulated. As Schnarrenberg is categorized as low
intensity residential (category 31) with an urban fraction of 0.5 and the UCM is applied here, energy is mainly
stored in the urban canopy layer instead of being transferred into the soil.
As this day was characterized by a shallow PBL and a temperature inversion, it is worth to investigate the PBL
evolution during the day. Figure 11 shows time-height cross sections of potential temperature at IPM (top) and
Schnarrenberg (bottom).
Both locations are characterized by a very stable shallow boundary layer until 09 UTC with a depth of less than
200 m. Between 03 and 09 UTC the temperatures at Schnarrenberg are up to 1.5 K colder near the surface (Fig. 9)
resulting in a stronger potential temperature gradient up to 400 m AGL compared to the IPM location. During the
day, the boundary layer height increases to 200--400 m above ground as indicated by the constant potential



temperature (e.g. Bauer et al., 2020) which is a typical value for European winter conditions (Seidel et al., 2012;
Wang et al., 2020). The PBL height estimates are confirmed by calculating the gradient Richardson number (Ri;
Chan, 2008) (not shown) which exceeds 0.25 at this altitude (Seidel et al., 2012; Lee and Wekker, 2016). After
sunset around 15:30 UTC the boundary layer collapses to a night-time stable boundary layer and a temperature
inversion occurs again.

### 4.2.  Air quality

The most relevant air pollutants for air quality considerations in cities are $NO_2$ and $PM_{10}$. Sources for these are
mainly truck supply, transit, and commuter traffic through the city as well as advection from motorways south,
west, and northwest of Stuttgart. We start with the discussion of the simulated horizontal distributions followed
by vertical cross sections of $NO_2$ and $PM_{10}$.

### 4.2.1 Horizontal distribution

Figure 12 shows the horizontal distribution of the $NO_2$ concentration at the lowest model half level (~15 m AGL)
at the four timesteps 07:30 UTC, 12 UTC, 18 UTC and 23 UTC 21 January 2019.
At 7:30 UTC the morning traffic rush hour is visible in the $NO_2$ concentrations in Fig. 12a. High $NO_2$
concentrations of more than 80 µg m$^{-3}$ are simulated along the motorway A81 in the northwest of the domain, over
the airport and over downtown Stuttgart. In the Neckar Valley the concentrations exceed 120 µg m$^{-3}$. At noon time
(Fig. 12b), when turbulence is fully evolved (Fig. 11), the simulated $NO_2$ concentrations are less than 30 µg m$^{-3}$
on average apparently due to vertical mixing of $NO_2$ (see next section). In the evening (Fig. 12c) the simulated
$NO_2$ concentrations increase again showing values of more than 100 µg m$^{-3}$ over the airport and more than 150 µg
m$^{-3}$ in downtown Stuttgart and the Neckar Valley due to road and air traffic. The high morning concentrations
along the northwestern motorway are not reached since the wind speed increases and the near surface winds turn
towards a westerly direction. According to the emission data set converted by the temporal factors, the evening
traffic spreads over a longer time.  During the night (Fig. 12d), $NO_2$ accumulates in the Stuttgart basin as well as
the Neckar Valley due to the very low nocturnal boundary layer height of less than 200 m capped by an atmospheric
inversion (Fig. 11).
Apart from $NO_2$, the concentration $PM_{10}$ is an important parameter for air quality considerations and is the decisive
factor for proclaiming a "fine dust alarm" situation in Stuttgart (Stuttgart Municipality and German Meteorological
Service (DWD), 2019). Note that the simulated $PM_{10}$ concentrations include particles less than 2.5 µm of diameter
($PM_{2.5}$) and that $PM_{10}$ is a diagnosed quantity in our model setup.
Figure 13 shows the horizontal distribution of $PM_{10}$ for the same time steps as shown in Fig 12.
During the morning traffic (Fig. 13a), $PM_{10}$ accumulates in the Stuttgart basin as this is an area with heavy traffic
during the morning and an atmospheric inversion is present (Fig. 7). Interestingly, the high $NO_2$ concentrations
along the motorway (Fig. 12a) do not lead to very high $PM_{10}$ concentrations potentially due to chemical transitions
caused by low temperatures.
During daytime when turbulence is fully evolved, the concentration of $PM_{10}$ decreases to less than 20 µg m$^{-3}$ due
to vertical mixing and horizontal transport (see next section). After sunset (Fig. 13c) PM10 starts to accumulate
again in the Stuttgart basin showing concentrations between 35—40 µg m$^{-3}$. During the night (Fig. 13d) $PM_{10}$
accumulates over a large part of the model domain as the nocturnal boundary layer is very shallow, an inversion
layer is present 400 m AGL and the wind direction changes from north to west.



### 4.2.2 Vertical distribution of NO$_2$ and PM$_{10}$

In addition to the horizontal distribution of near surface NO$_2$ and PM$_{10}$, TP simulations with a fine vertical resolution also enable qualitative insights into the vertical distribution of pollutants. Figure 14 shows West-East cross sections at Neckartor (Fig. 1b) during the morning rush hour and at noon time. Neckartor is one of the heaviest traffic locations in the Stuttgart city area.

The NO$_2$ concentration during the morning rush hour shows an accumulation along the motorway (red arrow in Fig. 14a) and in the region around Neckartor (white arrow in Fig. 14a) with concentrations exceeding 100 µg m$^{-3}$ as the atmospheric inversion prevents exchange with the layers above (Fig. 7). The vertical extent of concentrations higher than 30 µg m$^{-3}$ is about 200 m AGL with a strong reduction above.

During noon time (Fig. 14b), the simulated NO$_2$ concentration is much lower (less than 30 µg m$^{-3}$) as turbulence leads to a stronger mixing throughout the boundary layer up to 400 m AGL which is in accordance with the simulated potential temperature timeseries shown in Fig. 11.

Figure 15a displays the simulated PM$_{10}$ concentrations during the morning rush hour. Similar like for NO$_2$, higher concentrations of more than 25 µg m$^{-3}$ is simulated along the motorway and in the Stuttgart basin. During the day, PM$_{10}$ is vertically mixed showing a clear gradient around 800 m above sea level (ASL) (Fig. 15b) while concentrations remain between 10-20 µg m$^{-3}$ within the boundary layer.

Apart from the West-East cross sections it is also worthwhile to investigate the vertical temporal evolution of NO$_2$ and PM$_{10}$ concentrations. Therefore, Fig. 16 shows time height cross sections of NO$_2$ (top) and PM$_{10}$ (bottom) at Neckartor as an example for a heavy traffic area in the basin.

Well visible are the high simulated NO$_2$ and PM$_{10}$ concentrations during the morning rush hour with peak values of more than 120 µg m$^{-3}$ NO$_2$ and more than 40 µg m$^{-3}$ PM$_{10}$. The high concentrations of NO$_2$ and PM$_{10}$ are present up to around 150-200 m AGL. During daytime, turbulence efficiently mixes the pollutants up to higher altitude and the near surface concentrations are quickly reduced. During the evening when the very shallow boundary layer has developed again and evening traffic commences, the particle concentrations increase, and peak values of more than 30 µg m$^{-3}$ are simulated below 100 m AGL.

## 5. Summary and conclusion

This paper describes the setup of a AQFS prototype using WRF-Chem for the Stuttgart Metropolitan area. Because of the complex topography in this region, this simulation system requires a very high horizontal resolution down to the turbulence permitting scale to represent all orographic and land cover features.

For the development of this prototype 21 January 2019 served as test case as this was a typical winter day with an atmospheric inversion. In addition, this day was characterized as "fine dust alarm" situation where the PM$_{10}$ concentration at one of the heaviest traffic areas in the Stuttgart basin was expected to exceed 30 µg m$^{-3}$. The model setup encompassed three domains down to a turbulence permitting resolution of 50 m.

The initial conditions were provided by the ECMWF operational analysis, the CAMS reanalysis and WACCM model for background chemistry. Emission data sets from CAMS-REG-AP and high-resolution data with 500 m resolution from LUBW were combined to be used in the AQFS. As current emission data sets only provide annual totals or means, a temporal decomposition following TNO was applied (Denier van der Gon et al., 2011).





For this case study, we focused on the results with respect to 2-m temperature, surface fluxes and boundary layer
evolution as well as horizontal and vertical distributions of $NO_2$ and $PM_{10}$.
Our results revealed that despite the complex topography in Stuttgart, the model is in general capable to simulate
a realistic diurnal cycle of 2-m temperatures although, compared to observations, differences of up to 1 K occur.
Apparently the model has difficulties with the dissolution of low stratus clouds between 03 and 06 UTC  which
was also reported in the work of Steeneveld et al. (2015)  resulting in a warm 2-m temperature bias during the
morning. Although no measurements are available, the surface sensible heat fluxes show a clear diurnal cycle with
the magnitude clearly depending on the underlying land cover type. The low simulated ground heat flux and its
fluctuations between 00 UTC and sunrise partially confirm the fog dissolution issue but more test cases are needed
for a more detailed investigation. Over grid cells where the single layer UCM is active, most of the ground heat
flux is stored in the canopy layer thus not transferred into the soil. The high vertical resolution of 100 levels enables
a realistic representation of the nocturnal and daytime temperature inversion with an accompanying shallow
boundary layer of less than 400 m during the day.
The simulation of $PM_{10}$ shows an exceedance of the 30 µg m$^{-3}$ concentration threshold at the Neckartor station and
also fulfills the other fine dust alarm criteria shown in section 3.  Compared to the usually unevenly distributed air
quality measurements, the AQFS allows further insights into the spatio-temporal pollutant distribution. The
horizontal distributions of $NO_2$ and $PM_{10}$ at this particular day clearly indicate the main polluted areas along the
motorways and in the Stuttgart basin. The special orography of Stuttgart with its basin favors the accumulation of
$NO_2$ and $PM_{10}$ in the morning and evening while the pollutants are well mixed to around 200-400 m AGL when
the boundary layer is fully evolved.
The simulation also shows that pollutants can be advected from the motorway A81 towards Stuttgart, depending
on the wind situation potentially leading to an increase of the $NO_2$ and $PM_{10}$ concentrations in the Stuttgart basin.
As can be seen from Figs. 12 and 13, the Neckar Valley can also have a large impact on the pollutant concentration
in the Stuttgart basin in case an atmospheric inversion together with prevailing easterly winds is present.
This is, to our knowledge, the first study of applying WRF-Chem on a TP resolution for an urban area. To derive
more robust conclusions with respect to air pollution, more cases studies with different weather situations during
winter and summer time are necessary. Nevertheless, our evaluation gives the following indications to further
improve the quality of such simulations:
I.    Applying high spatial and temporal resolution gridded emission data from all pollution sources in near

real time to avoid extrapolating annual emissions to individual days.. This will help to enhance the

simulation of the diurnal cycles of chemical species.

II.    Improving the chemical background e.g. by applying higher resolution products from the CAMS

European Air quality project (Marécal et al., 2015). This will help to have a more detailed structure of

the chemical constituents beneficial for further downscaling simulations.

III.    Using a longer spin-up period and applying a larger TP model domain to further improve the spin-up of

turbulence in the model

IV.    Considering vertical distribution of surface emissions (e.g. Bieser et al., 2011; Guevara et al., 2020)
V.    Considerably increase the number of pollutant measurements to allow more robust conclusions



The AQFS has a great potential for urban planning applications. For example, land cover could be changed from
urban low density to urban high density to investigate the impact of urban re-densification e.g. on temperature and
air quality. Although no BEP can be applied on the TP resolution with our combination of parameterizations,
changes of the parameters required for the single layer UCM offer the opportunity to perform sensitivity analysis
with respect to different building heights, urban greening effects (Fallmann et al., 2016), or anthropogenic heat
(Karlický et al., 2020).
Although air quality modeling on the TP scale is a very challenging and computationally expensive task, we are
convinced that the AQFS will have a great potential to further improve process understanding and will certainly
help politicians to make decisions on a more scientifically valid basis.
**Code and data availability**
The WRF-Chem code version 4.0.3 can be downloaded from https://github.com/wrf-
model/WRF/archive/v4.0.3.tar.gz. ECMWF analysis data can be obtained from https://apps.ecmwf.int/archive-
catalogue/?type=an&class=od&stream=oper&expver=1 (last access: 26 August 2020). The user's affiliation needs
to belong to an ECMWF member state to benefit from these data sets. Due to restrictions on the input data sets for
this simulation, the data can only be made available upon special request from the corresponding author.
**Author Contributions**
TS prepared all emission data, set up the model and performed the simulation supported by HSB. HSB reclassified
the CORINE land use data set. KWS and TB conceived the idea and coordinated the project with VW. TS prepared
all figures and wrote the manuscript with input from all authors. All authors equally contributed to the scientific
discussion and helped to shape the research.
**Competing interests**
The authors declare that they have no conflict of interest.
**Acknowledgements**
This study has been performed within the EU-funded project *Open Forecast* (Action number 2017-DE-IA-0170).
We acknowledge ECMWF for providing analysis data from the operational IFS and CAMS reanalysis. The
Emissions of atmospheric Compounds and Compilation of Ancillary Data (ECCAD) system is acknowledged for
providing the CAMS-REG-AP Emission data set. We acknowledge the use of the WRF-Chem preprocessor tool
MOZBC, provided by the Atmospheric Chemistry Observations and Modeling Lab (ACOM) of NCAR. The
Baden-Württemberg State Institute for the Environment (LUBW) is highly acknowledged for providing high-
resolution annual emission data and for the high-resolution land cover data. Joachim Ingwersen from the
Department of Biogeophysics at the University of Hohenheim is acknowledged for converting the soil texture data.
Joachim Fallmann from the University of Mainz is acknowledged for providing the necessary code enhancement
of the dry deposition driver module to correctly couple the urban canopy model. The simulation was performed
on the national supercomputer Cray XC40 Hazel Hen at the High Performance Computing Center Stuttgart
(HLRS) within the WRFSCALE project..

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





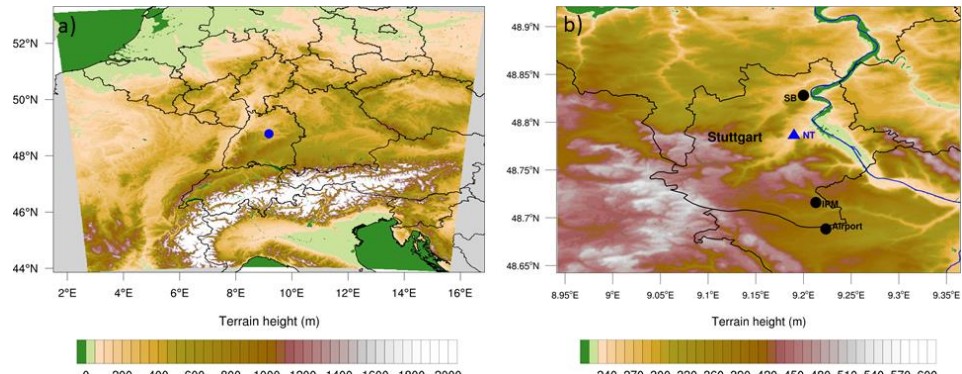

**Figure 1: Model domain 1 (a) and domain 3 (b). The blue dot in (a) denotes Stuttgart. Black dots in (b) show the location of the meteorological measurement sites. The blue diamond in (b) denotes the Neckartor (NT) location and the blue contour line denotes the Neckar River (River data © OpenStreetMap contributors 2020. Distributed under a Creative Commons BY-SA License).**





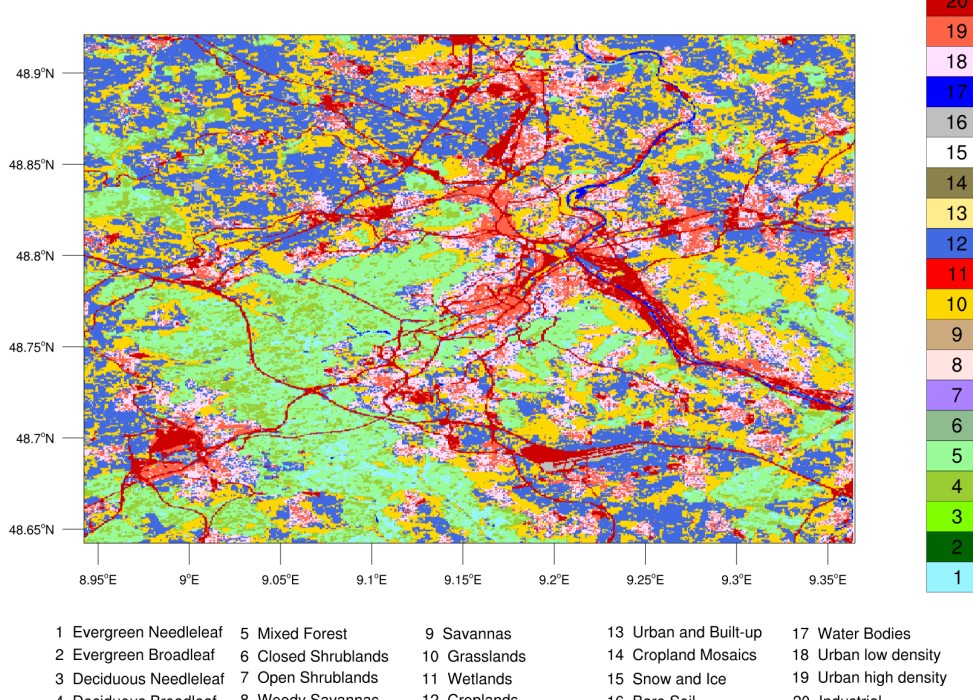

**Figure 2: Land cover data from the Baden-Württemberg State Institute for the Environment, Survey and Nature Conservation (LUBW) reclassified for WRF in the innermost domain at a resolution of 50 m.**





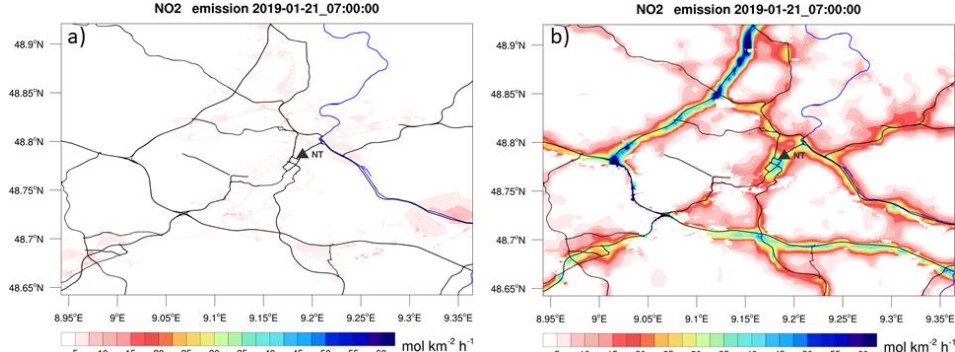

**Figure 3: NO₂ emissions valid at 07 UTC on January 21, 2019.** (a) shows the emissions derived from the CAMS-REG-AP data set and (b) shows the emissions derived from the BW-EMISS data set (Map Data © OpenStreetMap contributors 2020. Distributed under a Creative Commons BY-SA License).







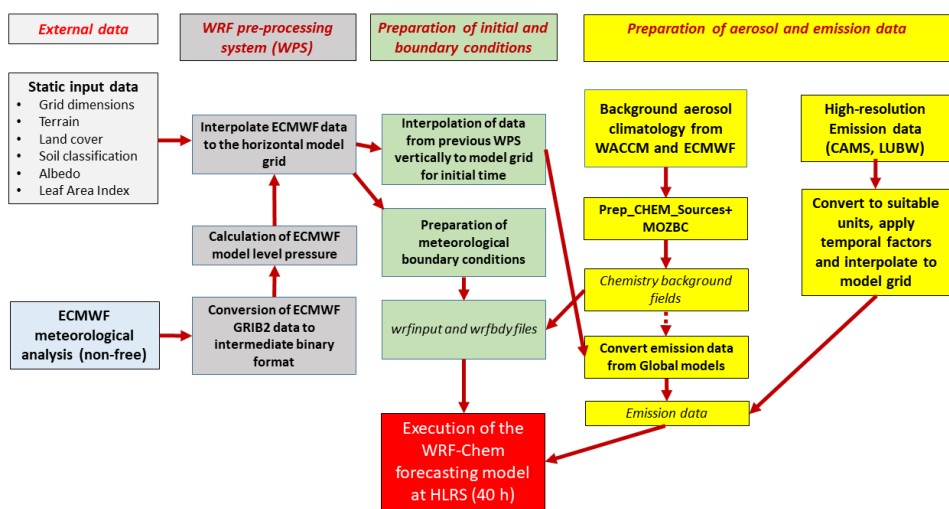

**Figure 4: Workflow of the AQFS prototype system.**





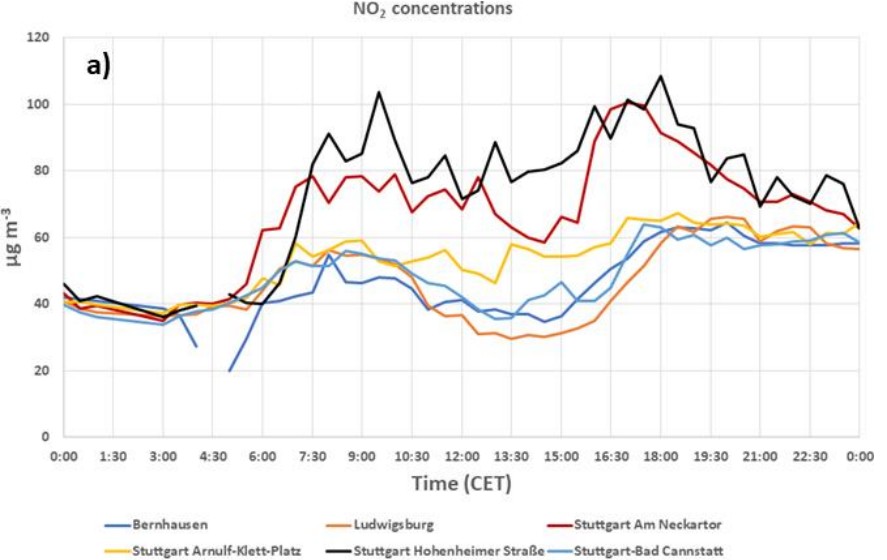

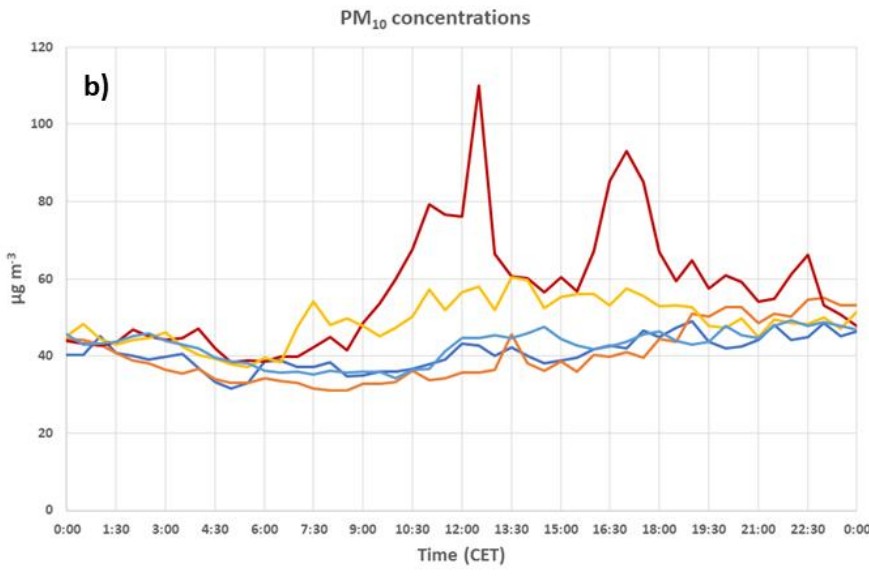

**Figure 5: Observed NO₂ (a) and PM₁₀ (b) concentrations at several stations distributed over the model domain on 21 January 2019. The time zone (CET) corresponds to local time. Measurements at Neckartor, Hohenheimer Strasse, and Arnulf-Klett Platz are directly taken next to the main road.**




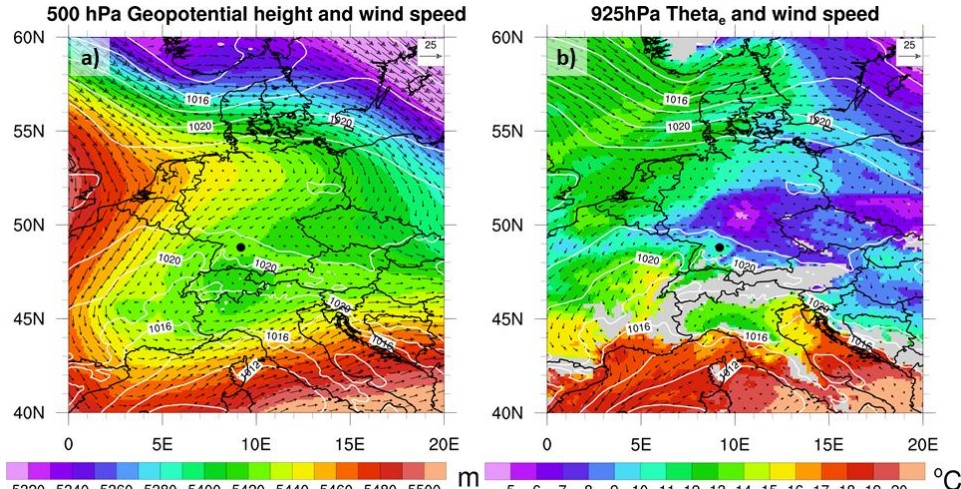

**Figure 6: (a) ECMWF operational analysis of 500 hPa geopotential height, sea level pressure (white contour lines) together with 500 hPa wind velocities valid at 00 UTC 21 January 2019. (b) shows the 925hPa equivalent potential temperature together with 925 hPa wind velocities and sea level pressure (white contour lines). Gray areas indicate values below the ECMWF model terrain. The black dot denotes Stuttgart and the reference wind vector length (top right corner of each Figure)) is equal to 25 m s$^{-1}$.**







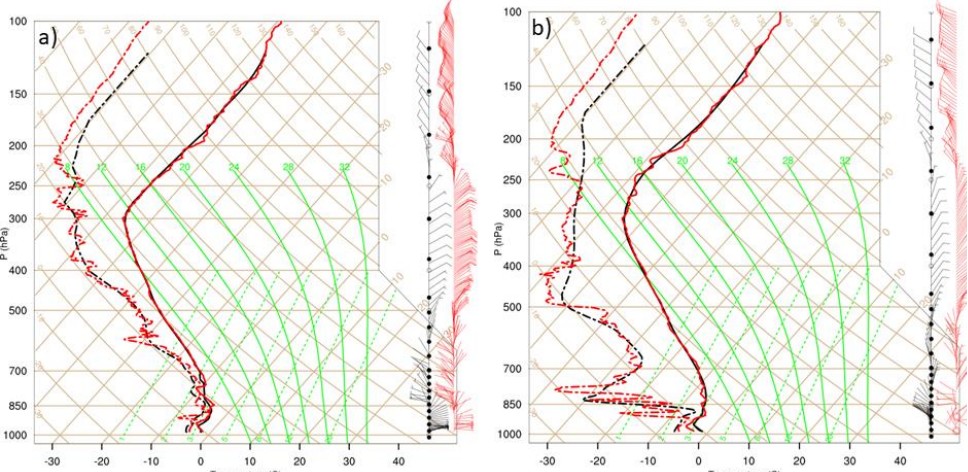

**Figure 7: Comparison of temperature, dewpoint and wind of the WRF model simulation (black line) and the sounding from Stuttgart-Schnarrenberg (red line) valid at 00 UTC (a) and 11 UTC (b) 21 January 2019.The solid lines denote the temperature profile and the dash-dotted line denotes the dewpoint profile. Wind barbs denote wind speed in m s⁻¹.**





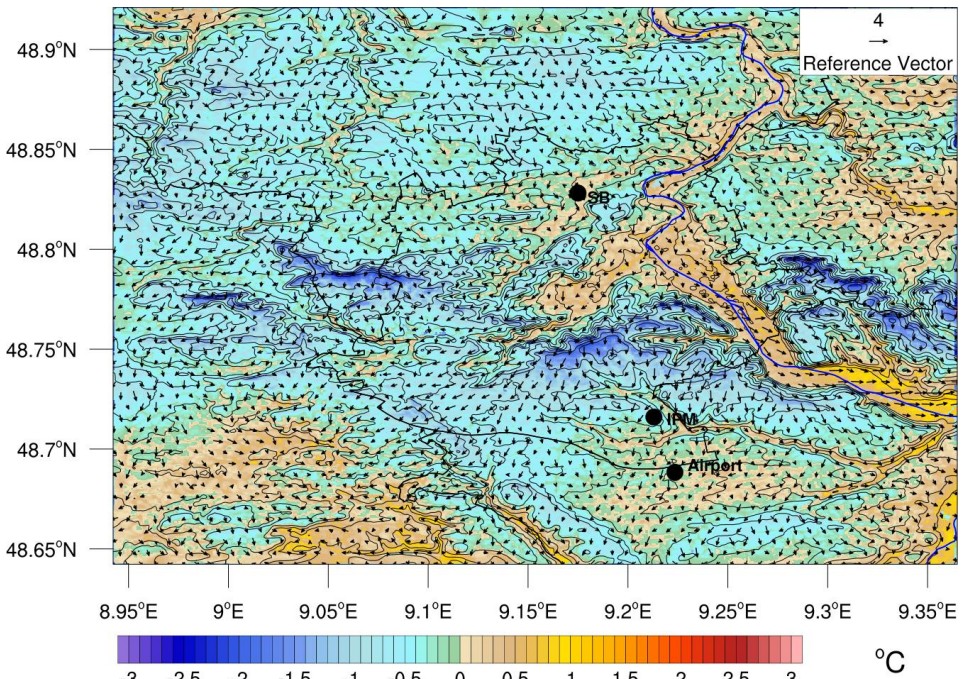

**Figure 8: 2-m temperature together with 10-m wind velocities at 12 UTC 21 January 2019. The thick black line denotes the Stuttgart city limits and the thin black contour lines denote the terrain. The blue line denotes the Neckar River (River data © OpenStreetMap contributors 2020. Distributed under a Creative Commons BY-SA License).**





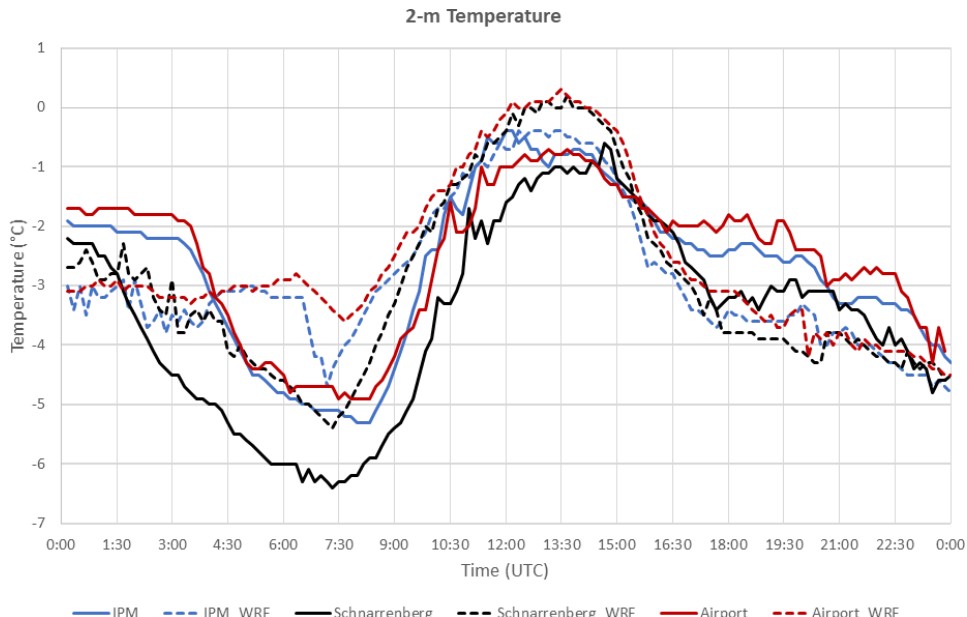

**Figure 9: Diurnal cycle of 2-m temperatures for the three meteorological stations shown in Fig. 3. Solid lines denote the observation, dashed lines denote the model simulation. The temporal resolution is 10 minutes.**







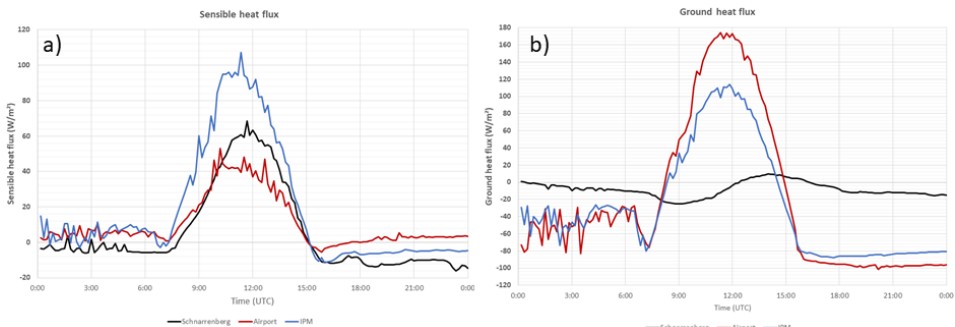

**Figure 10: Diurnal cycle of simulated sensible heat flux (SH, a) and ground heat flux (GRDFLX, b) at the three meteorological stations (white dots in Fig. 3). Positive values of GRDFLX indicate fluxes into the soil. The land cover categories are bare soil (airport), croplands (IPM), and urban (Schnarrenberg).**





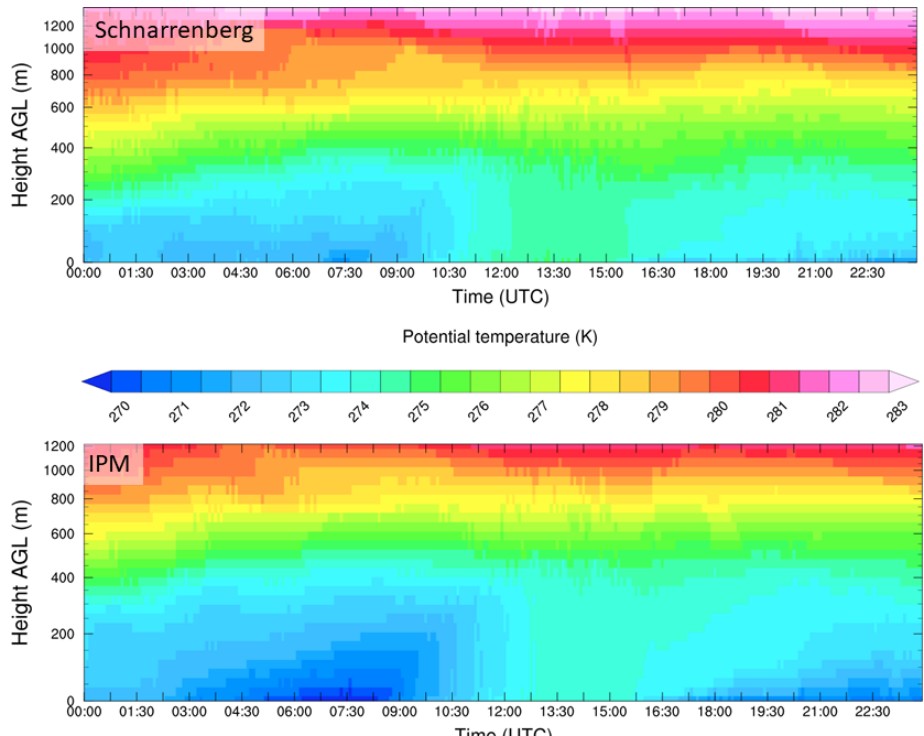

**Figure 11: Time-height cross section of the simulated potential temperature at IPM (top) and Schnarrenberg (bottom). The displayed altitude is above ground level (AGL).**







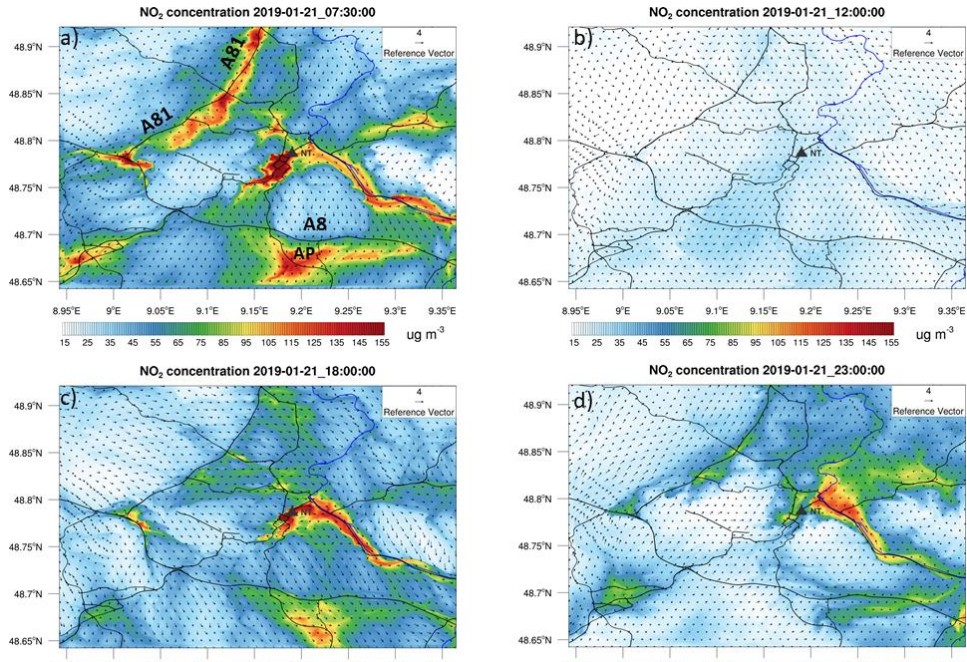

**Figure 12: NO₂ concentration at the lowest model level for 07:30 UTC, 12 UTC, 18:00 UTC, and 23 UTC (from a to d) 21 January 2019. The black contour lines denote main roads and motorways in and around Stuttgart (Map Data © OpenStreetMap contributors 2020. Distributed under a Creative Commons BY-SA License). AP denotes the airport, A8 and A81 denote the main motorways around Stuttgart.**





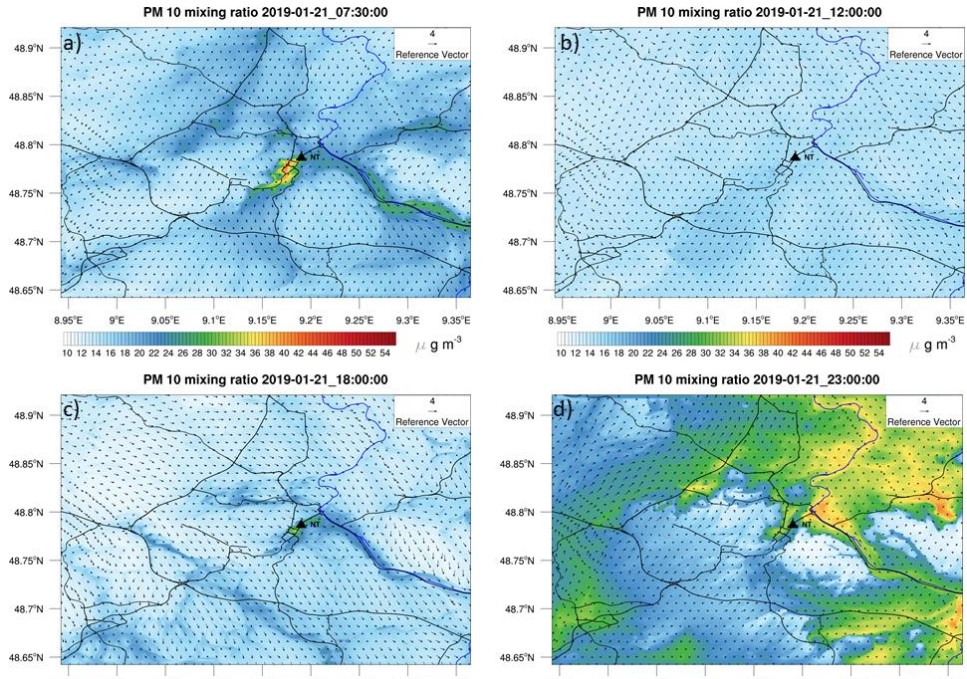

**Figure 13: Same as Fig. 12 but for PM₁₀ (Map Data © OpenStreetMap contributors 2020. Distributed under a Creative Commons BY-SA License).**





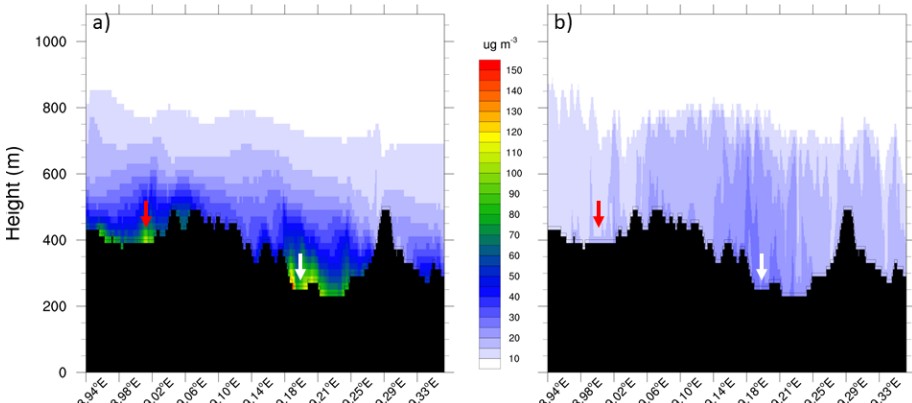

**Figure 14: West-East cross section through Neckartor displaying the NO₂ concentration at 07:30 UTC (a) and 12 UTC (b), 21 January 2019. The red arrow denotes the motorway A81 and the white arrow denotes the Neckartor location. The black area shows the model terrain above mean sea level.**







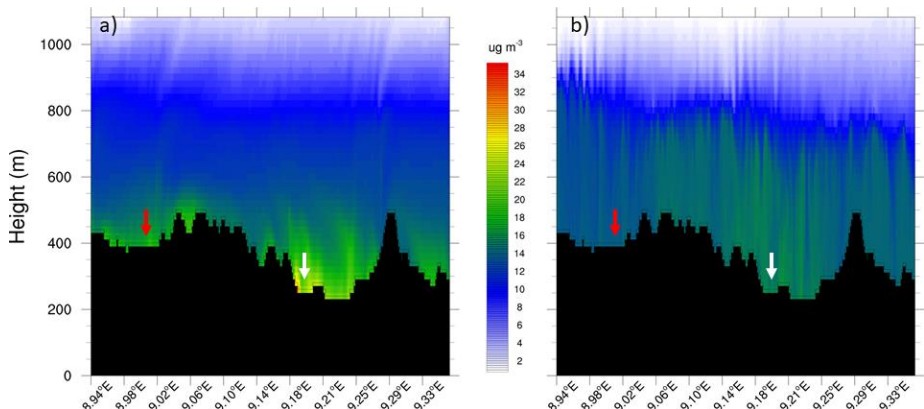

**Figure 15: Same as Fig. 14 but for PM₁₀.**




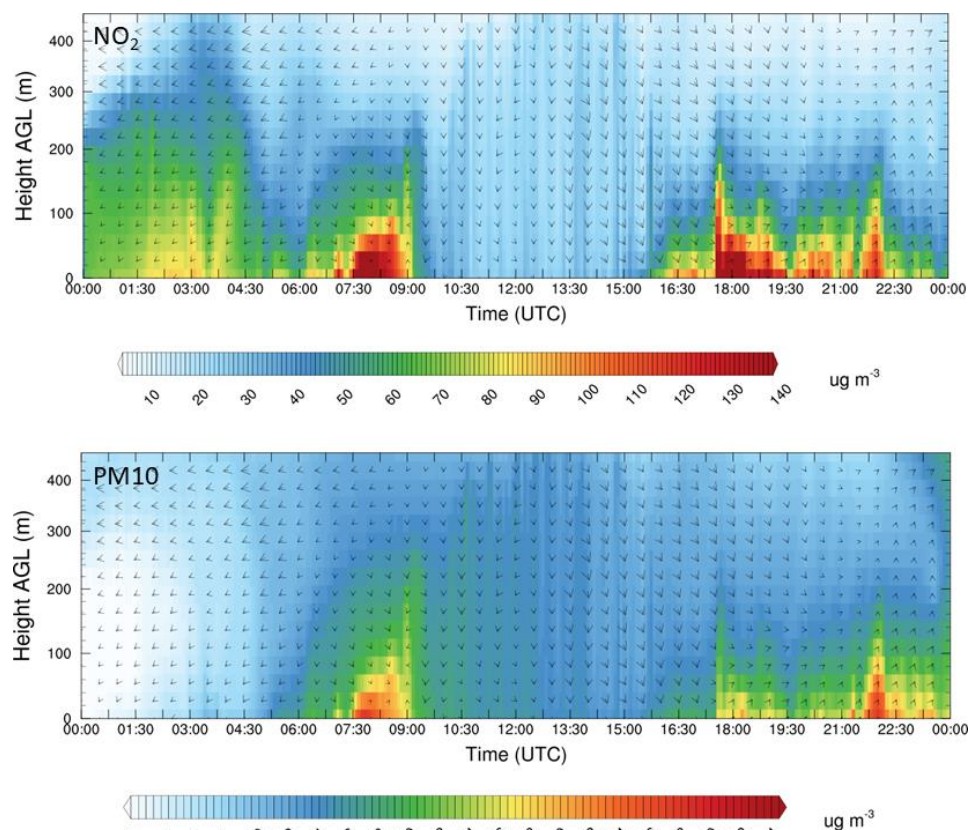

**Fig. 16: Time height cross section of NO₂ (top) and PM₁₀ (bottom) at Neckartor (NT) up to an altitude of 450 m AGL.**
