# Peer review of "Turbulence-permitting air pollution simulation for the 2 Stuttgart metropolitan area"

_Atmospheric Chemistry and Physics, 2020_

## Referee Comment (RC1) · Anonymous Referee #1 · 27 Nov 2020

The paper treats an interesting topic, is clearly structured and offers no significant language problems. With the paper being comparatively short, a lack of detail exists with regard to a more detailed presentation of the related processes, when it comes to the assessment of the interactions between meteorology and air chemistry. The introduction is relatively long compared to the most import aspects highlighted in the title of the paper which is air pollution modelling.

Referring to e.g. line 225-232, air quality has not been evaluated due to the temporal mismatch of emission data and modelling time. Quickly checking on the publicly available observations for the studied locations, I do find the model capable of representing actual conditions at least for NO2 and therefore would add a chemical evaluation respectively, for the sake of completeness. With regard to NO2, you even indicate an
overestimation of various peaks. That aspect will be addressed later.

Please further try to improve the statements on the added value compared to other studies existing for resolutions 1-3 km, for that particular area. For instance, past model approaches usually massively underestimated surface levels of pm10 due to various reasons. Due to the high resolution emission however, your system tends to improve that aspect. Please comment on this.

When using these kind of models – this has been mentioned in the introduction and conclusion – one might be interested in the forecast of pollution thresholds. Please comment on the point, how suitable that model system would be for actual applications, also comparing with other model systems with that purpose such as PALM-4U. Please provide more details on the added value of your system and also provide some insight on the computational costs, which might be an important information for a potential end-user. With that regard, please provide a synthesis of planned efforts, how that system could be transferred to operational use and if that is planned at all.

2.1 With decisions mostly being based on near surface concentration, how does the lowest model level of $\sim$15m addresses that aspect?

4. In order to get a more complete impression of the robustness of the model results for being used in urban areas, it would be interesting to include actual urban stations in the evaluation process.

Line 308: adding a central urban location would be interesting here

Figure 9: provide more details on the related processes here, especially on the reasons of the temporary increase @IPM and airport at about 4:30

Figure 11: The naming of the figures according to their location seems to be in the wrong order here. With the high vertical resolution being applied in the model, it would be interesting to see a comparable image for the observed potential temperature as well here.

Figure 12 nicely shows the potential of the high resolution, but a hint towards the observed quantity would be an added benefit.

Figure 13: While NO2 remains fairly static over the traffic areas, PM10 strongly accumulates in the north eastern part of the domain. Please discuss the reason for that.

Figure 14: highlight the cross section in one of the figures above. Further a large part of the figure is covered by the topography. Due to that, a lot of information gets lost for the most interesting areas in the lower urban boundary layer. Please modify the figure accordingly, to see what is going on under the arrows.

Figure 5: As mentioned earlier, it seems that the model is well capable of representing realistic conditions at least for the urban background. With regard to NO2 it even overestimates the peaks. Please add respective information

Line 405: What is the reference to this exceedance?

Line 424: unclear what exceedance you are referring to here.

431-434: That aspect is not clearly visible from the mentioned figures.

With regard to the project description ‘OpenForecast’: How open would that system be for local stakeholders and would it be capable to be used for actual decisions. Please briefly comment.

---

## Referee Comment (RC2) · Anonymous Referee #2 · 26 Dec 2020

The Stuttgart area is characterized by complex topography and known for frequent occurrence of high pollution concentrations. The paper describes a demonstration case study for this area with WRF-Chem at a horizontal resolution of 50 m for a polluted winter day. Overall, the paper addresses an important and interesting topic as air quality studies at such a high resolution are not frequently presented. The paper should definitely be published as an interesting case study with a high-resolution air quality model. However, it may be considered as a bit premature to regard the described setup as an 'air quality forecasting system (AQFS)'.

Specific comments

Abstract, line 26: 'Together with information about the vertical distribution of PM10 and $NO_2$ from the model, AQFS will serve ...' What is the difference between the

described WRF-Chem simulation and AQFS? Are there any special tools developed for the AQFS? If so, they should also be described in more detail in the paper.

The introduction is quite long considering the length of the paper and not all points mentioned in the introduction are relevant in the context of the paper (e.g. lines 29-36, remarks offline air quality models or on aerosol-radiation interactions).

Line 86: In the meantime, PALM offers also several option to use e.g. COSMO of WRF as drivers (Kadasch et al. and Lin et al. in https://gmd.copernicus.org/articles/special_issue999.html as well as the implementation of WRF_interface, which is already included in the PALM system). Besides of this, this remark would better match in the context of lines 96-100.

Line 158: The statement about RADM2 is somewhat odd (63 chemical species including photolysis and more than reactions). Besides oif this, the number of 63 includes also water vapor and a passive compound.

The authors claim that the model can be applied 'in a forecast and warning mode' (line 173). On the other hand, it is obvious from the domain size and the high resolution the simulations must be quite demanding, which is also mentioned in line 174. How high was the computational effort for the described study (How long did the simulation take on how many nodes)? Besides of this, tools should be supplied to stakeholders for an AQFS.

Lines 208-209: The differences should be commented in more quantitative way (E.g. what is the difference between the spatial integral for a) and b)?).

Section 2.4: Lines 225-232 are not related to observation and should be moved to somewhere else. Chemistry observations from Fig. 5 should also be included here instead.

Lines 291-307: It is not clearly mentioned whether too persistent simulated clouds are really the reason to the too late drop in temperature.

Line 345: Some comments on the influence of the quite thick lowest layer on the results should be given and (in spite of the objections and restrictions) a small comparison with observed values.

Line 357 'nocturnal boundary layer height': This is hardly visible in Fig. 11, either us different colors or indicate the PBL height in the figure.

Line 362 'that PM10 is a diagnosed quantity in our model setup.' What does this statement mean? I assume that Figures 12 and 13 show instantaneous values of the concentrations. What is the temporal variability at this high resolution? A presentation of the diurnal course of the pollutants should also be included. What is the effect of the too late drop in temperature on pollutant concentrations?

Figures 1, 2, 3, 8, 12 and 13: The lat-lon presentation used here is quite unfavorable.

Figure 4: Prep_chem_sources and MOZBC are separate tools, why are the put together? What is the meaning of the dashed arrow?

Figure 7: It is hard to distinguish any details. A presentation ut to a height of 400 hPa would be sufficient.

Figure 11: It is hard to distungish the PBL height from this figure, either change colors or indicate the PBL height.

---

## Author Comment (AC1) · 3 Feb 2021

We thank Referee #1 for his positive feedback and valuable comments. Please find our point by point responses below. Line numbers in our responses refer to the new manuscript without tracked changes.

The paper treats an interesting topic, is clearly structured and offers no significant language problems. With the paper being comparatively short, a lack of detail exists with regard to a more detailed presentation of the related processes, when it comes to the assessment of the interactions between meteorology and air chemistry. The introduction is relatively long compared to the most import aspects highlighted in the title of the paper which is air pollution modelling.

Referring to e.g. line 225-232, air quality has not been evaluated due to the temporal mismatch of emission data and modelling time. Quickly checking on the publicly available observations for the studied locations, I do find the model capable of representing actual conditions at least for NO2 and therefore would add a chemical evaluation respectively, for the sake of completeness. With regard to NO2, you even indicate an overestimation of various peaks. That aspect will be addressed later.

**Please see our comment with respect to your comment on Figure 12.**

Please further try to improve the statements on the added value compared to other studies existing for resolutions 1-3 km, for that particular area. For instance, past model approaches usually massively underestimated surface levels of pm10 due to various reasons. Due to the high-resolution emission however, your system tends to improve that aspect. Please comment on this.

Thank you for your comment. We tried to further emphasize the advantages of convection permitting simulations for our particular area. We added a new paragraph in the model set-up section 2.1 on page 5, line 166:

"Compared to a previous study from Fallmann et al. (2016), who performed simulations over the Stuttgart metropolitan area using WRF-Chem on a CP resolution of 3 km, or the study of Kuik et al. (2016) who performed a three month simulation at different resolutions over Berlin, simulations on the TP resolution provide a much more realistic representation of the land-cover structures (see Fig. 2 in this paper and e.g. Fig. 2b in Fallmann et al. (2016)). As the climate in the Stuttgart metropolitan area is strongly influenced by the topography, we are convinced that our special combination of a TP resolution and high-resolution emission data (see section 2.3) will lead to a better understanding and prediction of the air pollution situation in this area."

When using these kind of models – this has been mentioned in the introduction and conclusion – one might be interested in the forecast of pollution thresholds. Please comment on the point, how suitable that model system would be for actual applications, also comparing with other model systems with that purpose such as PALM-4U.

In our opinion there are several important points determining the quality of air pollution forecast models like WRF-Chem and PALM-4U: 1) Accurate initial conditions of the background chemistry, potentially also by means of (variational) data assimilation (e.g. Sun et al; 2020). Currently these background fields are only available on, compared to the CP or TP resolution, relatively coarser grid scales. However, this requires a major increase of the number of observations, and 2) a high spatial and temporal, near-real time emission data set which does not only contain traffic emission data but also emissions from the industrial sector.

Sun, W., Liu, Z., Chen, D., Zhao, P., and Chen, M.: Development and application of the WRFDA-Chem three-dimensional variational (3DVAR) system: aiming to improve air quality forecasting and diagnose model deficiencies, Atmos. Chem. Phys., 20, 9311–9329, https://doi.org/10.5194/acp-20-9311-2020, 2020.

Please provide more details on the added value of your system and also provide some insight on the computational costs, which might be an important information for a potential end-user.

The added value of our model system compared to previous existing studies using onlinecoupled atmosphere-chemistry models applied in major German or European cities is the TP resolution. To our knowledge, this was a novelty at the time our study was conducted. Our study provides a seamless approach from large atmospheric scales down to the TP resolution using high-resolution, although not real-time, emission data.

Depending on the actual weather situation, e.g. having a sophisticated cloud microphysics could be beneficial as warm rain schemes, as for example applied in PALM 6.0 have several limitations during convective events. We consider our AQFS using WRF-Chem as another useful tool to study air quality in certain areas.

As information about computational costs was also requested by reviewer #2, we added two paragraphs to the "experimental set-up" section on page 5, line 173 and it now reads:

"Currently, air pollution modeling with WRF-Chem is a computationally expensive task. Depending on the number of output variables and frequency (5 min in our study), a 24 h simulation currently takes around 36 h wall clock time. This is partly because the parallel NetCDF (PNetCDF) implementation in WRF is not very efficient for large files thus each file write takes between 20--25 s for a size of approx. 11 GB. For future experiments it is worth to try the I/O quilting option in combination with PNetCDF which should considerably reduce the time spent on I/O.

While the WRF model itself is ready for hybrid parallelism (MPI + OpenMP), the WRF-Chem model can only be used with MPI. If WRF-Chem could be enhanced for additional OpenMP capabilities, this would lead to an increase in computation speed almost linear with the number of OpenMP threads."

With that regard, please provide a synthesis of planned efforts, how that system could be transferred to operational use and if that is planned at all.

Within the OpenForecast project the idea was set up a prototype configuration of an AQFS for the Stuttgart metropolitan area. It was also planned to present our prototype to the Stuttgart municipality during the project period. However due to unexpected difficulties with the necessary input data set, the set-up of the prototype took longer than expected.

In a potential future project, the focus should be set on an improvement of the I/O (by means of quilting) and additional OpenMP capabilities but this still requires around 1500-2000 compute cores for operational use due to the small numerical time step. The high-resolution emission data sets will still remain a challenging task.

The following was added to the summary on page 14, line 491:

"In the future, more emphasis should also be put on an improvement of the I/O (e.g. by means of quilting) and additional OpenMP capabilities in WRF-Chem. However simulations with WRF-Chem at the TP resolution will still require around 1500-2000 compute cores for operational use due to the small numerical time step necessary."

**2.1 With decisions mostly being based on near surface concentration, how does the lowest model level of 15m addresses that aspect?**

Thank you for your comment. The terrain following coordinate system in the WRF model imposes a lower limit on the lowest model half level. Due to the required surface layer scheme, serving as a coupler between the land surface and the atmosphere, we cannot further reduce the layer thickness at this particular high resolution. Nevertheless, it is quite possible, that because of the terrain following coordinate system, some features may be even better represented as, e.g., compared to the PALM-4U model. PALM-4U applies a cartesian grid and thus, to our knowledge, does not consider slope effects e.g. in connection with radiation.

Unfortunately, we do not have 3-dimensional measurements available, but according to recent study of Samad et al. (2020) and an older study of Glaser et al. (2003), at least during daytime the concentrations of  $PM_{10}$  and  $NO_2$  are often almost constant up to an altitude of few 100 m above ground. Further studies are necessary for the stable nighttime PBL.

The paragraph on Page 10, line 360 was enhanced and now reads:

"As the incorporated emissions are from 2014 and are based on annual values, it cannot be expected that the model exactly matches the observed concentrations. For instance, the actual traffic, the sequence of traffic lights and traffic congestions of this particular day cannot be realistically represented. In addition, all diagnosed or prognostic chemical quantities are only available on model levels (with the lowest model half level being at ~15 m above ground) but according to studies of Glaser et al. (2003) and Samad et al. (2020) the concentrations of PM10 and NO2 are often constant up to 150—200 m AGL during daytime."

*Glaser, K., Vogt, U., Baumbach, G., Volz-Thomas, A., and Geiss, H. (2003), Vertical profiles of O3, NO2, NOx, VOC, and meteorological parameters during the Berlin Ozone Experiment (BERLIOZ) campaign, J. Geophys. Res., 108, 8253, doi:10.1029/2002JD002475, D4*

A. Samad, U. Vogt, A. Panta, D. Uprety: Vertical distribution of particulate matter, black carbon and ultra-fine particles in Stuttgart, Germany, Atmospheric Pollution Research, Volume 11, Issue 8, 2020, Pages 1441-1450, ISSN 1309-1042, https://doi.org/10.1016/j.apr.2020.05.017.

4. In order to get a more complete impression of the robustness of the model results for being used in urban areas, it would be interesting to include actual urban stations in the evaluation process.

Unfortunately, over the last few years, the number of pollution and meteorological measurement stations have been reduced so that no further urban stations are available in Stuttgart for evaluation.

**Line 308: adding a central urban location would be interesting here**

Thank you for your suggestion. We decided to add another line from Schlossplatz in Downtown Stuttgart (land use category 32, high density residential) to Fig. 10. Additional information with respect to the location "Schlossplatz" was added to the paragraphs on page 9, starting line 329 and on page 10, starting at line 340. Also, the location "Schlossplatz (SP)" was added to Fig. 1.

Figure 9: provide more details on the related processes here, especially on the reasons of the temporary increase @IPM and airport at about 4:30

We added two sentences on page 9, line 311 for clarification:

"A reason for this delayed temperature drop could be a simulated thin cloud layer around 1000 m AGL which is present in the lower left and partly the lower right quadrant of the model domain. This cloud layer slowly moves in a southeasterly direction and starts to dissolve around 06 UTC."

Figure 11: The naming of the figures according to their location seems to be in the wrong order here. With the high vertical resolution being applied in the model, it would be interesting to see a comparable image for the observed potential temperature as well here.

Thank you for detecting this. The figure caption itself is correct but the panel labeling was wrong. This has been corrected.

It would be indeed interesting to compare the simulated time series with observations but unfortunately no vertical profile measurements of potential temperature are available in the Stuttgart metropolitan area.

**Figure 12 nicely shows the potential of the high resolution, but a hint towards the observed quantity would be an added benefit.**

Thank you for your comment. We added the observed NO2 concentrations for the locations Neckartor, Hohenheimer Strasse (both in the city), and Bernhausen (next to the airport) to Figure 5a and the paragraph starting on page 11, line 383:

"Compared to the observed NO2 concentrations (Fig. 5a), the simulated concentrations during the peak traffic times are too high at Arnulf-Klett Platz, Neckartor and Hohenheimer Strasse. Possible reasons are that either the traffic is reduced and/or that the vehicle emission classification have been improved since 2014. Another contributing factor could be that the vertical mixing near the surface is too weak during sunrise and sunset while it appears slightly too strong during daytime as indicated by the very low simulated NO2 concentrations."

Figure 13: While NO2 remains fairly static over the traffic areas, PM10 strongly accumulates in the north eastern part of the domain. Please discuss the reason for that.

As the WRF-Chem code is very complex, we unfortunately cannot draw a robust conclusion here. We added the following paragraph to section 4.2.1, page 11, line 400:

"In the configuration we use in our study,  $PM_{10}$  is a diagnostic variable which is a sum of the  $PM_{2.5}$  concentration (which is around 26 µg m-3 at 23 UTC) and the other prognostic aerosol species. As the night is very cold with temperatures far below freezing and the humidity is very high, the high concentrations could imply a very (too) strong deposition or be the result of dense fog formation due to weak near-surface winds."

Figure 14: highlight the cross section in one of the figures above. Further a large part of the figure is covered by the topography. Due to that, a lot of information gets lost for the most interesting areas in the lower urban boundary layer. Please modify the figure accordingly, to see what is going on under the arrows.

Thank you for your suggestion. We highlighted the cross section in Fig. 13a by a red line and also changed the vertical extent of the panels in Figs. 14 and 15 to highlight what is going on near the surface.

Figure 5: As mentioned earlier, it seems that the model is well capable of representing realistic conditions at least for the urban background. With regard to NO2 it even overestimates the peaks. Please add respective information

Thank you for your suggestion. We added the information about the overestimation on page 11, line 383. See also our response in connection to your comment on Fig. 12.

**Line 405: What is the reference to this exceedance?**

The reference to this exceedance is the LUBW measurement. We slightly modified the sentence on page 12, line 438 to include a reference: "In addition, this day was characterized as "fine dust alarm" situation where the  $PM_{10}$  concentration at the station Neckartor in the Stuttgart basin was expected to exceed 30 µg m-3 (http://www.stadtklima-

stuttgart.de/stadtklima\_filestorage/download/luft/Feinstaubwerte-2019\_AN.pdf)."

Line 424: unclear what exceedance you are referring to here.

Thank you for your comment. We checked our results again. The model did not exceed the  $30 \ \mu g \ m^{-3}$  directly at the Neckartor measurement location but a few grid cells next to it did. The sentence on page 13, line 457 is changed to:

"The simulation of  $PM_{10}$  shows an exceedance of the 30 µg m-3 concentration threshold very close to the Neckartor station and also fulfills the other fine dust alarm criteria shown in section 3."

**431-434: That aspect is not clearly visible from the mentioned figures.**

Unfortunately this situation cannot be fully explained by the single images shown in Figs. 12 and 13. We therefore decided to provide animations of  $NO_2$  and  $PM_{10}$  as supplementary material to underline this aspect.

With regard to the project description 'OpenForecast': How open would that system be for local stakeholders and would it be capable to be used for actual decisions. Please briefly comment.

Currently, there are several limitations of our prototype with respect to an operational AQFS.

First, the high-quality operational ECMWF analysis data on model levels is not publicly available without charges for other purposes than research.

Secondly, it is still very difficult to obtain near real-time emission data on a very high spatial and temporal resolution. More and more traffic emission data sets become available but to our knowledge, they are not yet available in near-real time.

Thirdly, the WRF-Chem model is not yet ready to use for hybrid parallelism (MPI + OpenMP) which limits the number of compute cores for the simulation. If a future version of WRF-Chem would support hybrid parallelism, this will considerably speed up the simulation by at least a factor 2-3 compared to the current situation.

If the above-mentioned limitations could be overcome, the AQFS will be ready and available for decision making.

---

## Author Comment (AC2) · 3 Feb 2021

We thank Referee #2 for his positive feedback and valuable comments. Please find our point by point responses below. Line numbers in our responses refer to the new manuscript without tracked changes.

The Stuttgart area is characterized by complex topography and known for frequent occurrence of high pollution concentrations. The paper describes a demonstration case study for this area with WRF-Chem at a horizontal resolution of 50 m for a polluted winter day. Overall, the paper addresses an important and interesting topic as air quality studies at such a high resolution are not frequently presented. The paper should definitely be published as an interesting case study with a high-resolution air quality model. However, it may be considered as a bit premature to regard the described setup as an 'air quality forecasting system (AQFS)'.

The results presented in our manuscript are the first steps towards an AQFS. We emphasized that this is still a prototype which, by including additional refinements/enhancements, has the potential to be applied as a forecast system in the future.

Specific comments

Abstract, line 26: 'Together with information about the vertical distribution of PM10 and NO2 from the model, AQFS will serve . . .' What is the difference between the described WRF-Chem simulation and AQFS? Are there any special tools developed for the AQFS? If so, they should also be described in more detail in the paper.

Our AQFS utilizes the WRF-Chem model including the improvements and enhancements described in sections 2.1-2.3.

The introduction is quite long considering the length of the paper and not all points mentioned in the introduction are relevant in the context of the paper (e.g. lines 29-36, remarks offline air quality models or on aerosol-radiation interactions).

We consider it as essential to provide the reader with information about pollutions limits proposed by the World Health Organization (WHO) as this is the basis for the current discussions of protection scenarios. From our point of view, it is also important to mention that there is a longer history of offline models which are presently still widely used around the globe and that the direct interaction between radiation and aerosols is essential. Following your suggestion, we slightly shortened the introduction and the 2nd and 3rd paragraph now read:

"Due to a strong increase of road traffic in major European cities (Thunis et al., 2017), pollution limits are often violated in larger cities. E.g. for particulate matter with particle diameters less than 10 μm (PM10), the critical value is an annual mean concentration of 20 μg m$^{-3}$ or a daily mean value of 50 μg m$^{-3}$ (WHO, 2005). For Nitrogen dioxide (NO$_2$) the critical values are 200 μg m$^{-3}$ and 40 μg m$^{-3}$ as daily and annual mean values, respectively.

The violation of these pollution limits can lead to health and environmental problems and is currently part of several litigations e.g. at the German Federal Administrative Court dealing with possible driving bans for non low-emission vehicles. The basis for these litigations are mostly few, unevenly distributed local observations. In combination with special meteorological conditions like winter time thermal inversion layers it can be misleading to conclude about the overall air quality in the city only from single observations. According to e.g. the German Federal Immission Control Ordinance it is sufficient that traffic related measurements are representative for a section of 100 m, but this is not representative for the commercial and office districts in the cities that are suffering from traffic control in case of fine dust alerts and residential areas. Namely in residential areas health protection action plans require representative air quality measures."

Line 86: In the meantime, PALM offers also several option to use e.g. COSMO of WRF as drivers (Kadasch et al. and Lin et al. in https://gmd.copernicus.org/articles/special_issue999.html as well as the implementation of WRF interface, which is already included in the PALM system). Besides of this, this remark would better match in the context of lines 96-100.

At the time when the simulation was conducted, these driver options were not available to the community. As our focus is on the WRF model system, we decided to add the work of Lin et al. (2020) to the outlook on page 14, line 488:

"Recently, Lin et al. (2020) developed an interface to use output from high-resolution WRF simulations to force PALM 6.0 in an offline mode which could be another tool in the future to study microscale structures in urban areas."

Line 158: The statement about RADM2 is somewhat odd (63 chemical species including photolysis and more than reactions). Besides of this, the number of 63 includes also water vapor and a passive compound.

Thank you for pointing this out. In this sentence, the number of chemical reactions was missing. To avoid confusion on the exact number of species used in RADM in WRF-chem, we decided to change the sentence on page 5, line 158 to:

"RADM2 features more than 60 chemical species and more than 135 chemical reactions including photolysis"

The authors claim that the model can be applied 'in a forecast and warning mode' (line 173). On the other hand, it is obvious from the domain size and the high resolution the simulations must be quite demanding, which is also mentioned in line 174. How high was the computational effort for the described study (How long did the simulation take on how many nodes)? Besides of this, tools should be supplied to stakeholders for an AQFS.

Thank you for your comment. As also reviewer #1 asked for more information, we added the following paragraphs to the "experimental set-up" section on page 5, line 173:

"Currently, air pollution modeling with WRF-Chem is a computationally expensive task. Depending on the number of output variables and frequency (5 min in our study), a 24 h simulation currently takes around 36 h wall clock time. For future experiments it is worth to try the I/O quilting option in combination with PNetCDF which should considerably reduce the time spent on I/O.

While the WRF model itself is ready for hybrid parallelism (MPI + OpenMP), the WRF-Chem model can only be used with MPI. If WRF-Chem could be enhanced for additional OpenMP capabilities, this would lead to an increase in computation speed almost linear with the number of OpenMP threads."

We are not sure what is meant by "tools" here? If the AQFS is further refined in the future, this will require a short tutorial for the person or institution operating this system including all pre- and postprocessing steps.

Lines 208-209: The differences should be commented in more quantitative way (E.g. what is the difference between the spatial integral for a) and b)?).

Thank you for your comment. We decided to add the average emission rates from both data sources. The average emission for the CAMS-REG-AP data set is 2 mol $km^{-2} h^{-1}$ and the average emission for the BW-EMISS data set is 7 mol $km^{-2} h^{-1}$.
The following sentence was added on page 7, line 226:

"The average emissions for this particular time step are 2 mol $km^{-2} h^{-1}$ for the CAMS-REG-AP data set and 7 mol $km^{-2} h^{-1}$ for the BW-EMISS data set."

Section 2.4: Lines 225-232 are not related to observation and should be moved to somewhere else. Chemistry observations from Fig. 5 should also be included here instead.

Thank you for your suggestion. This paragraph has been moved to section 4.2 and has been slightly modified. It now reads (page 10, line 360):

"As the incorporated emissions are from 2014 and are based on annual values, it cannot be expected that the model exactly matches the observed concentrations. For instance, the actual traffic, the sequence of traffic lights and traffic congestions of this particular day cannot be realistically represented. In addition, all diagnosed or prognostic chemical quantities are only available on model levels (with the lowest model half level being at ~15 m above ground) but according to studies of (Glaser, 2003) and (Samad et al., 2020) the concentrations of $PM_{10}$ and $NO_2$ are often constant up to 150—200 m AGL during daytime."

Following the suggestion of reviewer #1 we also included the simulated $NO_2$ concentrations in Fig. 5. Figure 5 is also referenced in section 4.2.1 on page 11, line 383.

*Glaser, K., Vogt, U., Baumbach, G., Volz-Thomas, A., and Geiss, H. (2003), Vertical profiles of O3, NO2, NOx, VOC, and meteorological parameters during the Berlin Ozone Experiment (BERLIOZ) campaign, J. Geophys. Res., 108, 8253, doi:10.1029/2002JD002475, D4*

*A. Samad, U. Vogt, A. Panta, D. Uprety,*
*Vertical distribution of particulate matter, black carbon and ultra-fine particles in Stuttgart, Germany, Atmospheric Pollution Research, Volume 11, Issue 8, 2020, Pages 1441-1450, ISSN 1309-1042, https://doi.org/10.1016/j.apr.2020.05.017.*

Lines 291-307: It is not clearly mentioned whether too persistent simulated clouds are really the reason to the too late drop in temperature.

We checked the model output with respect to hydrometeors again. The model simulates a very thin cloud layer at around 1000 m above ground level along the lower left and partly the lower right quadrant of the model domain. This thin cloud layer slowly moves to the SE and starts to diminish around 6 UTC.

We added two sentences on page 9, line 311 for clarification:

"A reason for this delayed temperature drop could be a simulated thin cloud layer around 1000 m AGL which is present in the lower left and partly the lower right quadrant of the model domain. This cloud layer slowly moves in a southeasterly direction and starts to dissolve around 06 UTC."

Line 345: Some comments on the influence of the quite thick lowest layer on the results should be given and (in spite of the objections and restrictions) a small comparison with observed values.

With respect to the observations, please see our comment on section 2.4 above. Unfortunately we do not have 3-dimensional measurements available, but according to a recent study of Samad et al. (2020) and an older study of Glaser et al. (2003), at least the concentrations of $PM_{10}$ and $NO_2$ are often constant up to an altitude of a few 100 m above ground. We added a paragraph on page 10, line 360:

"As the incorporated emissions are from 2014 and are based on annual values, it cannot be expected that the model exactly matches the observed concentrations. For instance, the actual traffic, the sequence of traffic lights and traffic congestions of this particular day cannot be realistically represented. In addition, all diagnosed or prognostic chemical quantities are only available on model levels  (with the lowest model half level being at ~15 m above ground). According to studies of Glaser et al. (2003) and Samad et al. (2020) the concentrations of $PM_{10}$ and $NO_2$ are often constant up to 150—200 m AGL during daytime."

Line 357 'nocturnal boundary layer height': This is hardly visible in Fig. 11, either use different colors or indicate the PBL height in the figure.

We decided to add the potential temperature gradient to the original Fig. 11. The PBL height can be determined by the strongest gradient (dark blue colors).

The paragraph starting on page 10, line 343 now reads:
"Both locations are characterized by a very stable shallow boundary layer until 09 UTC with a depth of less than 200 m. Between 03 and 09 UTC the temperatures at Schnarrenberg are up to 1.5 K colder near the surface (Fig. 9) resulting in a stronger potential temperature gradient up to 400 m AGL compared to the IPM location. During the day, the boundary layer height increases to 400 m above ground as indicated by the constant potential temperature (e.g. Bauer et al., 2020) which is a typical value for European winter conditions (Seidel et al., 2012; Wang et al., 2020). The PBL heights are also visible by the potential temperature gradients ($\Delta\theta$) shown in Figs. 11c, d. During the morning hours, a very shallow boundary layer was simulated at Schnarrenberg (blue colors in Fig. 11c) while at IPM some fluctuations are present. During daytime, $\Delta\theta$ nicely shows the PBL height evolution up to 400 m AGL, while after sunset the PBL collapses to a very stable layer again (dark blue colors in Figs. 11c, d) with heights between 50—100 m AGL. Calculating the gradient Richardson number (Ri; Chan, 2008) (not shown) and assuming a threshold of 0.25 for a turbulent PBL 0.25 (Seidel et al., 2012; Lee and Wekker, 2016) gives similar results. After sunset around 15:30 UTC, the boundary layer collapses to a night-time stable boundary layer and a temperature inversion occurred again. "

Line 362 'that PM10 is a diagnosed quantity in our model setup.' What does this statement mean? I assume that Figures 12 and 13 show instantaneous values of the concentrations. What is the temporal variability at this high resolution? A presentation of the diurnal course of the pollutants should also be included. What is the effect of the too late drop in temperature on pollutant concentrations?

Our intention was to explain that $PM_{10}$ is a sum of species, which have a diameter of less than 10 µm. All variables/fields shown in the paper are instantaneous values. We deleted this sentence from section 4.2.1 to avoid confusion.

To further illustrate the temporal evolution of $NO_2$ and $PM_{10}$, we decided to provide animations of the simulated $NO_2$ and $PM_{10}$ concentrations as supplementary material.

Regarding the effect of the too late temperature drop on the pollutant concentrations, we unfortunately cannot draw a conclusion here.

Figures 1, 2, 3, 8, 12 and 13: The lat-lon presentation used here is quite unfavorable.

We do not understand what is "unfavorable" here? The lat-lon labeling with geographical coordinates in decimal degrees is reasonable for us.

Figure 4: Prep_chem_sources and MOZBC are separate tools, why are the put together? What is the meaning of the dashed arrow?

We slightly modified the workflow diagram (Fig. 4) . The dashed arrow was introduced unintentionally.

Figure 7: It is hard to distinguish any details. A presentation up to a height of 400 hPa would be sufficient.

We changed the layout of the Skew-T diagrams shown in Fig. 7 by limiting the vertical extent to 400 hPa and reducing the temperature range in addition.

Figure 11: It is hard to distinguish the PBL height from this figure, either change colors or indicate the PBL height.

Please see our answer on your comment to line 357 and Figure 11 above.

---

## Referee Report (RR1)

Dear authors,

Thank you for revising the manuscript according to the comments. As also mentioned by reviewer2, I do see the premature status as a core problem for really assessing the model's performance for operational use. With other model activities are emerging in that area, e.g. in the framework of PALM-4U developments, it will be interesting however to know, how these results will compare with future studies/systems. A common problem might be, that these systems are way too complex for using it operationally, by 'external users'. You are giving some hints about potential modifications which still have to be accomplished, a detailed view on that is still not provided - or easy to do at this stage. I do think however, that the study could provide a starting point for future model intercomparisons and directions towards operational use.

A few things still pop up:

- It does not come out clearly, what will be needed to run this model in 'forecasting mode'. I expect the analysis presented refers to a hindcast mode?

-Figure 13: Could the reason for high PM values in the north west could be related to roads?

- the new Figure 5 is too busy, maybe divided in subplots

Chapter 4.2. The evaluation should come more as an introduction or more prominent, rather than being just a brief sentence in the paragraph.

- maybe not for direct discussion in the paper, it would be interesting if the model system WRF-Chem itself would be further maintained in the future

- The studies added to comment on the vertical distributions (Line 378-383) are not perfectly suited here, as one analysis summer conditions, and the other a more rural location. maybe you can find another study, highlighting that effect. Or provide a comment that with a 50x50m grid cell size, the height of 15m would be reasonable.